# Text2Lip: Progressive Lip-Synced Talking Face Generation from Text via Viseme-Guided Rendering

## Abstract

Generating semantically coherent and visually accurate talking faces requires bridging the gap between linguistic meaning and facial articulation. Although audio-driven methods remain prevalent, their reliance on high-quality paired audio visual data and the inherent ambiguity in mapping acoustics to lip motion pose significant challenges in terms of scalability and robustness. To address these issues, we propose **Text2Lip**, a viseme-centric framework that constructs an interpretable phonetic-visual bridge by embedding textual input into structured viseme sequences. These mid-level units serve as a linguistically grounded prior for lip motion prediction. Furthermore, we design a progressive viseme-audio replacement strategy based on curriculum learning, enabling the model to gradually transition from real audio to pseudo-audio reconstructed from enhanced viseme features. This allows for robust generation in both audio-present and audio-free scenarios. Finally, a landmark-guided renderer synthesizes photorealistic facial videos with accurate lip synchronization. Extensive evaluations show that Text2Lip outperforms existing approaches in semantic fidelity, visual realism, and modality robustness, establishing a new paradigm for controllable and flexible talking face generation.

## 1 Introduction

Talking face generation aims to synthesize photorealistic facial videos synchronized with speech or textual content, enabling applications in virtual avatars, remote education, and human-computer interaction. Most existing approaches adopt an *audio-driven* paradigm, where acoustic features directly guide lip motion via 3D deformable models Xu et al. (2024b); Wei et al. (2024) or facial landmarks Zhou et al. (2020); Wang et al. (2024). While audio provides rich temporal and prosodic cues for driving lip motion, obtaining high-quality paired audio-video data remains costly and error-prone. In contrast, text as input is more flexible and accessible, making it an attractive alternative, especially in low-resource or privacy-sensitive scenarios.

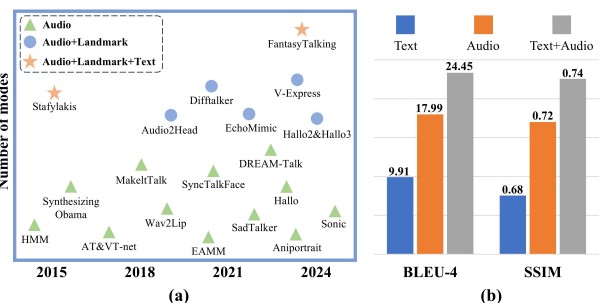

Figure 1: Modality usage in talking face generation. (a) Existing methods predominantly rely on audio inputs to drive lip motion. However, aligned audio-visual data is often scarce or costly to obtain. (b) Quantitative comparisons show that audio outperforms text-only input in fidelity, while combining both modalities yields further improvements. These observations motivate our viseme-centric approach, which leverages the accessibility and structure of text to bridge linguistic semantics and facial motion, enabling robust synthesis even in the absence of audio.

Apart from this, directly relying on audio overlooks the intrinsic coupling among *text*, *phonetics*, and *visual articulation* that underlies human communication. This leads to a critical limitation: speech-driven models often learn ambiguous mappings from audio to lip shapes which is the inherent many-to-one mapping between phonemes and visual articulation. For example, /p/, /b/, /m/ all

corresponding to the same bilabial closure viseme. This ambiguity compromises semantic precision and visual expressiveness, particularly when emotional clarity or linguistic fidelity is essential. As illustrated in Figure 1, these challenges motivate us to rethink modality design. We explore how the inherent structure of language, when grounded through phonetic and visemic representations, can effectively guide facial animation, even in the absence of audio.

To address this, we introduce **Text2Lip**, a viseme-aware framework that explicitly models the linguistic-phonetic-visual hierarchy. Instead of relying solely on audio, we convert input text into interpretable viseme sequences via word-to-phoneme-to-viseme mapping, which serve as semantically grounded priors for facial motion synthesis. This design disambiguates visually similar phonemes and enhances semantic-lip alignment.

Built upon this foundation, we develop a curriculum-based *Progressive Viseme-Audio Replacement* strategy. During training, we gradually transition from real audio to text-derived viseme embeddings, enabling our model to reconstruct pseudo-audio features through cross-modal attention. This approach not only allows seamless adaptation to audio-free settings but also enhances robustness against noisy or missing speech. To ensure high-quality visual synthesis, we adopt a landmark-guided *Photorealistic Video Rendering* module adapted from EchoMimic Chen et al. (2025), which converts the predicted landmarks and optional (real or pseudo) audio into temporally smooth, lip-synced talking face videos. Our contributions are summarized as follows:

- We propose a novel viseme-centric framework that systematically aligns linguistic, phonetic, and visual representations, thereby enabling more interpretable, controllable, flexible, and semantically consistent talking face video generation.
- We introduce a curriculum-based viseme-audio replacement strategy that facilitates flexible modality handling, supporting both audio-driven and audio-free generation scenarios.
- Our approach achieves state-of-the-art performance in lip synchronization, semantic consistency, and cross-modal robustness, demonstrated on multiple benchmarks under varied conditions.

## 2 RELATED WORK

### 2.1 AUDIO-DRIVEN TALKING FACE GENERATION

Audio-driven talking face generation is a long-standing core focus in audiovisual synthesis. Traditional methods employ explicit 3D representations, such as 3DMM-based approaches Zhang et al. (2023b;a); Wei et al. (2024); Xu et al. (2024b), which offer controllability but struggle to capture fine-grained facial dynamics. End-to-end models Chen et al. (2025); Xu et al. (2024a); Ferdowsifard et al. (2021) bypass 3D intermediates by directly mapping audio to pixels, improving realism and synchronization. Extensions like Hallo2/3 Cui et al. (2025a;b) enrich expression diversity via semantic prompts, but still rely on reference images for identity consistency. To enhance expressiveness, multimodal methods incorporate landmarks Wang et al. (2024), motion fields Wang et al. (2021), or textual cues Wei et al. (2025) to supplement audio. Despite progress, audio remains vulnerable to noise and lacks strong alignment with non-verbal expressions. These limitations motivate a shift toward text-driven paradigms. Our work explores a viseme-aware generation framework that reconstructs expressive lip motion from text alone, bypassing the limitations of unreliable audio input while preserving semantic-visual alignment.

### 2.2 LANDMARK-BASED LIP READING

Lip reading deciphers speech from silent video using visual cues. Early methods rely on hand-crafted features Lucey et al. (2008); Chan (2001); Luettin & Thacker (1997), while deep learning approaches Stafylakis & Tzimiropoulos (2017) improve spatiotemporal modeling via CNNs. However, most focus on recognition rather than generation. Lip-synchronized face generation reverses the task by synthesizing facial motion from audio. Models like Wav2Lip Prajwal et al. (2020), EAMM Ji et al. (2022), and SyncTalkFace Park et al. (2022) leverage lip landmarks or keypoints as intermediates to align audio and visual dynamics. Despite their success, these methods suffer from phoneme-to-lip ambiguity, where different phonemes share similar lip shapes, limiting generation fidelity. To address this, we introduce visemes—visually discriminative units of speech—as a

semantic-visual bridge. By modeling viseme-aware structures, our approach enhances disambiguation and improves expressiveness in text- or audio-driven facial animation.

## 3 METHODOLOGY

Although audio-driven talking face generation has made notable progress, it suffers from intrinsic audio-to-lip ambiguity: different phrases (*e.g.*, 'bad boy' vs. 'bat boat') can share nearly identical lip shapes, leading to semantically inconsistent or visually blurred outputs. This ambiguity limits both expressiveness and training stability, especially when semantic precision is needed. To address this, we introduce visemes, visual units that abstract away acoustic differences while capturing shared articulatory patterns, as an intermediate representation. Compared to phonemes, visemes offer a more visually grounded and interpretable signal to guide the movement of the lips. This structured visual prior improves semantic alignment, supports generalization to audio-free settings, and improves generation quality. We therefore propose a viseme-aware pipeline that transforms text into viseme sequences to guide subsequent lip motion synthesis. The detailed process is presented below.

### 3.1 VISEME-CENTRIC TEXT ENCODING

Accurate lip-reading generation hinges on capturing the visual manifestation of speech articulation. In this context, *visemes*, the minimal visual units representing distinct lip movements, serve as a theoretical and practical bridge between phonetic content and facial motion. Unlike phonemes, visemes abstract away acoustic variations such as vocal cord vibration, and instead focus on articulatory similarities. For instance, while /b/ and /p/ differ phonetically, both correspond to the visual pattern of lips suddenly parting after closure. This visual commonality positions visemes as a crucial intermediate representation for aligning text semantics with lip dynamics.

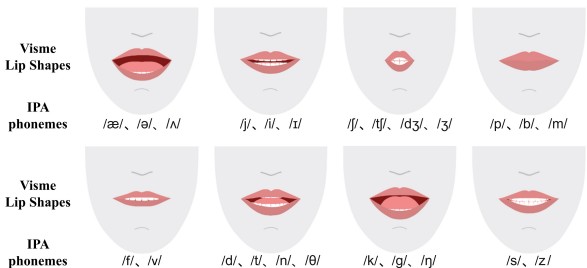

Figure 2: The correspondence between Viseme lip shapes and International Phonetic Alphabet (IPA) phonemes. Each row shows a set of typical lip shapes, and the IPA phonemes that trigger the lip shape are marked below.

To visually illustrate the phoneme-to-viseme abstraction, Figure 2 presents a representative mapping between International Phonetic Alphabet (IPA) symbols and their corresponding lip shapes. Each row depicts a typical viseme category, showcasing the shared articulatory appearance among phonemes within the same group. For example, for the phonemes /s/ and /z/, despite differing in vocal cord vibration, exhibit similar tongue and lip configurations, and are thus assigned to the same viseme. In contrast, visually distinct phonemes such as /a:/ and /i/ are grouped into separate viseme classes to preserve critical visual differences.

Building on this insight, we transform input text into fine-grained viseme sequences via a multi-stage pipeline. Specifically, we convert text into IPA sequences using established text-to-phoneme tools, with a dictionary-based fallback for common words to ensure semantic fidelity. Next, a predefined phoneme-to-viseme mapping table (from Microsoft's Speech API) is used to cluster phonemes into viseme categories based on articulatory similarity.

The resulting viseme sequence **V** encapsulates the visual articulation structure inherent in the text and serves as an interpretable input for downstream generation. It ensures that lip movements are not only visually plausible but also semantically coherent at the pronunciation level, forming a strong visual prior for accurate and expressive lip-reading synthesis.

### 3.2 PROGRESSIVE VISEME-AUDIO REPLACEMENT

While the viseme sequence **V** provides structured and semantically grounded visual instructions, directly generating temporally consistent and expressive facial landmarks from visemes alone remains challenging—especially under real-world conditions where audio is noisy, incomplete, or entirely

Figure 3: Overview of the proposed **Text2Lip**. It contains three key stages: (1) *Viseme-Centric Text Encoding* converts text to visemes via word-phoneme-viseme mapping, (2) *Progressive Viseme-Audio Replacement* adopts text-derived viseme embeddings to radually replace the audio using curriculum learning, and (3) *Photorealistic Landmark Rendering* synthesizes high-quality videos with precise lip synchronization via an adaptive EchoMimic renderer. The inference stage achieves realistic talking face videos driven solely by text.

absent. Most prior methods rely heavily on audio input as the temporal driving force, which limits their robustness and generalizability. To address this, we propose *Progressive Viseme-Audio Replacement* (PVAR), a curriculum learning strategy that progressively replaces audio features with viseme-derived from textual embeddings during training. This allows the model to gradually adapt to text-only inputs, while retaining audio guidance in early stages to stabilize training. Furthermore, we introduce a pseudo-audio reconstruction module to recover latent speech information from enhanced text features, ensuring high-fidelity landmark generation even without explicit audio.

**Viseme Feature Extraction.** We employ a Transformer-based multimodal encoder to extract high-level semantic features from viseme sequences. Each viseme token is first linearly projected into an embedding space with positional encoding:

$$v'_n = W^v \cdot v_n + b^v + PE_v(n), \tag{1}$$

where $v_n$ is the one-hot encoding of the $n$-th viseme in vocabulary $\mathcal{V}$, $W^v$ and $b^v$ are learnable parameters, and $PE_v(\cdot)$ denotes sinusoidal position encoding Vaswani et al. (2017). The resulting embedded sequence $\{v'_n\}_{n=1}^N$ is processed by a viseme encoder:

$$\tilde{v}_{1:N} = \text{VisemeEncoder}(v'_{1:N}). \tag{2}$$

**Landmark Generation with Modality Replacement.** To generate accurate facial landmarks under varying modality availability, we design a Transformer-based module that progressively drops the audio stream during training and reconstructs it via pseudo-audio generation. For each time step, facial landmarks and MFCC audio features are encoded as:

$$l'_m = W^l \cdot l_m + b^l + PE_l(m); a'_n = W^a \cdot a_n + b^a + PE_a(n), \tag{3}$$

where $l_m$ is the $m$-th landmark coordinate, $a_n$ is the MFCC audio feature, and $W^l, W^a, b^l, b^a$ are trainable weights and biases.

We introduce an audio dropout mechanism governed by a Bernoulli mask $\mathbf{M}$:

$$\tilde{a}_{1:N} = \mathbf{M} \odot \text{AudioEncoder}(a'_{1:N}), \mathbf{M} \sim B(1 - p_{\text{drop}}), \tag{4}$$

where $p_{\text{drop}}$ is linearly increased over training steps:

$$p_{\text{drop}} = p_{\text{start}} + \frac{(p_{\text{end}} - p_{\text{start}}) \cdot t}{T}, \tag{5}$$

with $t$ denoting the current step and $T$ the total steps. We set $p_{\text{start}} = 0$, $p_{\text{end}} = 1$ to ensure a full transition to text-only input.

**Viseme-Driven Audio Hallucination.** To compensate for missing audio, we enhance the viseme features using a gated feedforward unit:

$$\tilde{v}_{1:N}^{\text{enh}} = \text{LayerNorm}(\text{GLU}(\tilde{v}_{1:N})). \tag{6}$$

These enhanced features are fed into a cross-modal pseudo-audio generator based on multi-head attention:

$$\hat{a}_{1:N} = \text{MultiheadAttention}(\tilde{a}_{1:N}, \tilde{v}_{1:N}^{\text{enh}}, \tilde{v}_{1:N}^{\text{enh}}). \tag{7}$$

when $p_{\text{drop}}$ is equal to 1, $\tilde{a}_{1:N}$ is initialized to a blank tensor $Q \in \mathbb{R}^{B \times N \times d}$. The reconstructed pseudo-audio $\hat{a}_{1:N}$ is then fused through linear projection and gated modulation:

$$\hat{a}_{1:N}^{\text{pse}} = \hat{a}_{1:N} + \sigma(W_2 \phi(W_1 \cdot \text{Mean}(\hat{a}_{1:N}))) \cdot \hat{a}_{1:N}, \tag{8}$$

where $\text{Mean}(\cdot)$ is temporal average pooling over the sequence dimension, $W_1$ and $W_2$ are learnable projection matrices, $\phi(\cdot)$ is ReLU activation, and $\sigma(\cdot)$ is a sigmoid gating function.

**Cross-Modal Landmark Prediction.** Finally, we use a Transformer-based *LipDecoder* to synthesize facial landmark sequences by integrating pseudo-audio, viseme features, and previously generated poses:

$$\tilde{l}_{m+1} = \text{LipDecoder}(l'_{1:m}, \tilde{v}_{1:N}, \hat{a}_{1:N}^{\text{pse}}), \tag{9}$$

which can be decomposed into:

$$z_m = \text{CrossAttention}(l'_{1:m}, \hat{a}_{1:N}^{\text{pse}}, \hat{a}_{1:N}^{\text{pse}}); \tilde{l}_{m+1} = \text{CrossAttention}(z_{1:m}, \tilde{v}_{1:N}, \tilde{v}_{1:N}). \tag{10}$$

This design enables the model to synthesize temporally coherent and semantically aligned facial motion, even under modality-incomplete conditions.

## 3.3 Photorealistic Landmark Rendering

Having obtained temporally coherent and semantically aligned facial landmark sequences $\{\tilde{l}_m\}_{m=1}^M$ through PVAR-based compensation, we proceed to synthesize photorealistic talking face videos. This stage translates structural motion cues into pixel-level facial dynamics while preserving lip-audio synchronization and visual realism.

To achieve this, we adopt the state-of-the-art *EchoMimic* model Chen et al. (2025) as the video rendering backend. EchoMimic is a reference-based video synthesis framework that generates high-fidelity lip-synced talking faces by taking in a sequence of facial landmarks and auxiliary audio features. It employs a spatiotemporal consistency module to enforce smooth lip transitions and a texture completion mechanism to preserve identity and image quality.

Given the predicted landmark trajectory $\{\tilde{l}_{m+1}\}$ and optional audio features (real or pseudo), the video generation process is defined as:

$$\text{Video} = \text{Synthesis}(\{\tilde{l}_{m+1}\}_{m=1}^M, \text{audio}), \tag{11}$$

where audio can refer to original audio or reconstructed pseudo-audio, depending on availability. In scenarios where audio input is missing, our framework remains functional by relying solely on viseme-derived pseudo-audio features, ensuring robustness in low-resource or silent input conditions.

This design completes our pipeline from textual input to fully generated talking face video, enabling expressive, semantically accurate, and modality-flexible synthesis.

## 3.4 Training Protocol

We finally streamline our core framework as a three-stage design for text-driven lip-synced talking face generation, as illustrated in Figure 3.

**Stage I: Viseme-Centric Text Encoding** converts input text into a sequence of visual speech units using word-to-phoneme-to-viseme mapping, establishing a semantically aligned prior for lip motion.

**Stage II: Progressive Viseme-Audio Replacement** employs curriculum learning to gradually replace audio inputs with text-derived viseme embeddings. Cross-modal attention reconstructs pseudo-audio features from enhanced viseme representations, which are fused with visual cues to predict temporally coherent facial landmarks.

**Stage III: Photorealistic Landmark Rendering** synthesizes high-fidelity videos from landmarks and optional audio (real or pseudo-acoustic) using our adapted EchoMimic renderer, ensuring accurate lip synchronization and temporal smoothness.

This comprehensive framework demonstrates robust performance across different input modalities, supporting both audio-conditioned and text-only generation scenarios while maintaining high visual quality and articulation accuracy. The progressive three-stage design effectively bridges the semantic gap between linguistic inputs and visual outputs through carefully designed intermediate representations.

## 4 EXPERIMENTS

### 4.1 EXPERIMENTAL SETTINGS

**Datasets.** Following existing works Wang et al. (2021); Niu & Mak (2023); Wu et al. (2023); Li et al. (2025); El Ogri et al. (2024), we conduct experiments on well-established and widely adopted benchmarks **GRID** Cooke et al. (2006) and **AVDigits** Hu et al. (2016) in this research domain to evaluate the effectiveness of the proposed method. GRID contains videos of 33 speakers, covering a total of 51 words. Each sentence consists of six words, with a fixed sentence structure of command + color + preposition + letter + number + adverb (*e.g.*, lay red with y two again). All videos have a consistent duration of 75 frames. AVDigits captures the speech video of digits 0 to 9 spoken nine times by six speakers. Each video is recorded at 25 fps and the audio is recorded at 48 kHz. To improve data quality, we screen the original videos, removing abnormal samples such as those with excessive head movements, multiple faces, and inconsistencies in the speaker's voice. The processed data are divided into training, validation, and test sets according to Assael et al. (2016). In addition, we also self-collected data for open-domain evaluation. The self-collected open-domain dataset was primarily used for qualitative demonstration (as in Figure 5), aiming to visually showcase the model's capability in handling free-form text.

**Analysis on Dataset Selection.** Our decision to use these datasets is strategic: their controlled settings (*i.e.*, featuring single scenarios and frontal views) allow us to minimize confounding factors such as variable backgrounds, camera angles and recording environments. This focused setup enables a precise evaluation of the core capability of the text-driven paradigm to replace audio in guiding lip-synchronized video generation. While these datasets have a constrained vocabulary, this very limitation provides an ideal testbed for validating the feasibility and effectiveness of the novel cross-modal mapping from text to visemes to visual articulation that we propose. Experiment results demonstrate that even within this bounded lexicon, our text-driven approach achieves performance comparable to audio-driven methods, thereby establishing a solid foundation for future exploration on larger-scale, open-domain datasets.

**Evaluation Metrics.** We use multiple indicators to perform quantitative analysis from different dimensions. First, we use the **SSIM** Wang et al. (2004), **PSNR** Hore & Ziou (2010), **LPIPS** Zhang et al. (2018) and **FID** Heusel et al. (2017) to measure the visual similarity between the generated videos and the real videos at the image and video levels. **DTW-P** Sakoe & Chiba (2003) and **MPJPE** Ionescu et al. (2013) are adopted to measure the matching degree between the generated landmark sequence with ground-truth. Besides, we introduce the SyncNet Chung & Zisserman (2016) model to calculate **Sync-C** and **Sync-D** which Sync-C measures the synchronization consistency between lip sounds and audio and Sync-D evaluates the temporal consistency of dynamic lip movements. In addition, we design an inverse evaluation to measure the semantic preservation accuracy of generated videos. Specifically, we retrain the NSLT Camgoz et al. (2018) model, which interprets text sentences from lip reading videos and calculates the text alignment accuracy, such as **BLEU** Papineni et al. (2002), **WER** Assael et al. (2016), **DTW-P** Sakoe & Chiba (2003). In all experiments, when evaluating Sync-C and Sync-D, we consistently use the "real audio" provided in the dataset to calculate scores, never using "pseudo-audio," thus ensuring fairness to all methods.

**Data Prepossessing.** We adopt the DLib Kazemi & Sullivan (2014) library to accurately extract the 2D coordinates of 68 facial landmarks from the original videos. The extracted coordinates are globally normalized to eliminate individual and environmental variations, producing standardized 2D facial landmark labels. For audio data, we use the Mel-frequency cepstral coefficients (MFCC) to preprocess the original audio data, laying a foundation for subsequent model training and analysis.

Table 1: Quantitative comparisons with the state-of-the-arts on GRID and AVDigits datasets. ✓ indicates the modality used. **T** and **A** represent Text and Audio respectively.

| Methods | Modality | | GRID | | | | | | AVDigits | | | | | |
| | T | A | SSIM↑ | PSNR↑ | LPIPS↓ | FID↓ | Sync-C↑ | Sync-D↓ | SSIM↑ | PSNR↑ | LPIPS↓ | FID↓ | Sync-C↑ | Sync-D↓ |
|---|---|---|---|---|---|---|---|---|---|---|---|---|---|---|
| V-Express (arXiv 2024) | | ✓ | 0.687 | 16.693 | 0.283 | 115.701 | 4.170 | 8.429 | 0.719 | 15.723 | 0.277 | 176.767 | 3.850 | 6.445 |
| AniPortrait (arXiv 2024) | | ✓ | 0.698 | 17.573 | 0.279 | 48.722 | 2.019 | 11.907 | 0.713 | 18.932 | 0.252 | 88.403 | 1.349 | 9.046 |
| EchoMimic (AAAI 2024) | | ✓ | 0.728 | 18.059 | 0.274 | 38.798 | 3.687 | 8.820 | 0.728 | 19.128 | 0.241 | 61.237 | 3.692 | 6.764 |
| Hallo2 (ICLR 2025) | ✓ | ✓ | 0.733 | 17.532 | 0.291 | 69.896 | 4.170 | 8.299 | 0.735 | 19.355 | 0.247 | 57.625 | 3.694 | 6.490 |
| SadTalker (CVPR 2023) | | ✓ | 0.721 | 18.372 | 0.297 | 195.381 | 5.661 | 7.403 | 0.717 | 18.663 | 0.250 | 154.211 | 3.198 | **5.742** |
| Sonic (CVPR 2025) | | ✓ | 0.737 | 18.821 | 0.241 | 34.128 | **5.851** | **7.038** | 0.739 | 19.621 | 0.240 | 55.118 | **3.892** | 6.317 |
| Text2Lip (Ours) | ✓ | | **0.740** | **19.023** | **0.238** | **32.109** | 4.641 | 7.302 | **0.741** | **19.721** | **0.238** | **51.282** | 3.715 | 6.207 |

**Model Settings.** Our Text2Lip model is built from 2-layer, 4-head Transformers with a uniform embedding size of 512. All parts of the network are trained using the Adam optimizer with a learning rate of $1 \times 10^{-3}$ and a batch size of $batchsize = 128$. All experiments are conducted on a server equipped with 8 NVIDIA A40 GPUs.

## 4.2 COMPARISON WITH STATE-OF-THE-ARTS

We compare the Text2Lip with some mainstream talkface generation methods: V-Express Wang et al. (2024), AniPortrait Wei et al. (2024), EchoMimic Chen et al. (2025), Hallo2 Cui et al. (2025a), SadTalker Zhang et al. (2023b) and Sonic Ji et al. (2025). It is worth noting that all baseline models (including EchoMimic) are evaluated using their original architectures where audio serves as the mandatory core input signal, while text inputs remain optional auxiliary modalities. To ensure a fair comparison of single-modality driving capabilities, the reported baseline results correspond to configurations using only audio input. This establishes a direct and meaningful comparison with our Text2Lip approach, which operates exclusively on text input.

**Comparison on GRID.** As shown in Table 1, our method Text2Lip achieves superior or comparable performance across all major evaluation metrics, demonstrating the effectiveness of our audio-free design. In terms of visual quality, our model achieves the highest SSIM of 0.740 and PSNR of 19.023, demonstrating superior structural similarity and pixel-level fidelity compared to methods leveraging audio (*e.g.*, Sonic: SSIM 0.737, PSNR 18.821). Furthermore, we achieve the lowest LPIPS of 0.238 and FID of 32.109, highlighting enhanced perceptual realism and distributional closeness to real videos. While under this inherent "disadvantage" of lacking crucial prior information, Text2Lip successfully learns robust phoneme-viseme mappings and natural speech rhythm patterns implicitly from data through our sophisticated viseme-guided mapping and progressive training strategy, still achieving Sync-C 4.641 and Sync-D 7.302 competitive performance, surpassing V-Express and EchoMimic and approaching the performance of strong audio-based methods SadTalker and Sonic, demonstrating the feasibility and advantages of audio-free speaking face generation in real-world scenarios.

**Comparison on AVDigits.** As shown in Table 1, Text2Lip also demonstrates highly competitive performance on the AVDigits dataset. In terms of lip synchronization, Text2Lip achieves Sync-C scores of 3.715 and Sync-D scores of 6.207, comparable to or exceeding audio-based methods such as SadTalker (3.198/6.317) and V-Express (3.850/6.445), demonstrating its ability to ensure lip sync coherence without audio supervision. These results validate the feasibility of the Text2Lip text-only framework for generating natural, synchronized speaking faces and extend its utility to scenarios with limited or no audio.

**Comparison on Landmark.** Both lip landmark generation and sign language generation face the core challenge of mapping discrete linguistic symbols to continuous human motion trajectories. Despite their different application scenarios, both require solving the fundamental problem of language-to-motion mapping Therefore, we further evaluate Text2Lip landmark generation capabilities on the GRID dataset using reliable baselines in the

Table 2: Landmark comparison results on GRID dataset.

| Methods | BLEU-1↑ | BLEU-4↑ | WER↓ | DTW-P↓ | MPJPE↓ |
|---|---|---|---|---|---|
| PT-GN (ECCV 2020) | 40.87 | 9.76 | 52.17 | 4.97 | 807.49 |
| GEN-OBT (MM 2022) | 36.36 | 6.35 | 56.33 | 6.55 | **761.35** |
| LVMCN (ICASSP 2025) | 47.85 | 15.36 | 45.41 | 5.32 | 807.37 |
| Text2Lip(Ours) | **54.81** | **23.50** | **39.43** | **3.27** | 783.32 |

sign language domain, such as PT-GN Saunders et al. (2020), GEN-OBT Tang et al. (2022), and LVMCN Wang et al. (2025). The results are shown in Table 2. Compared to the three methods, Text2Lip performs better in terms of semantic fidelity of landmark sequences. Notably, its WER is 39.43, significantly lower than all the compared methods, further confirming the accuracy of its landmark sequence reconstruction. Although our method slightly underperformed GEN-OBT on MPJPE, it achieved a balance between semantic accuracy and geometric precision overall. As a geometric metric, semantic correctness is far more critical than millimeter-level coordinate accuracy in talking face generation. This is analogous to a lip shape being perceived as "correct" (high semantic quality) being far more valuable than one that numerically matches a reference frame (low MPJPE) but is unintelligible.

**User Study.** We conduct subjective evaluations on GRID and AVDigits datasets across four key dimensions: naturalness, visual clarity, temporal consistency, and smoothness. A total of 100 participants rated the results of the five compared methods on a scale of 1 to 5. As

Table 3: User study comparison on GRID dataset.

| Methods | Naturalness | Visual clarity | Temporal consistency | Smoothness |
|---|---|---|---|---|
| Sonic | 3.35 | 3.16 | 3.22 | 3.05 |
| Hallo2 | 2.82 | 2.92 | 2.71 | 2.75 |
| AniPortrait | 2.57 | 3.01 | 2.63 | 2.71 |
| V-Express | 1.62 | 2.19 | 2.31 | 2.28 |
| Text2Lip (Ours) | **3.97 (19%↑)** | **4.23 (34%↑)** | **4.26 (32%↑)** | **4.19 (37%↑)** |

shown in Table 3, our Text2Lip method outperformed the other methods across all four dimensions, particularly achieving a significant 37% improvement in smoothness.

## 4.3 ABLATION STUDY

**The Role of Viseme-centric Text Encoding.** The experimental analysis in Table 4 shows that viseme-centric encoding effectively bridges the semantic gap, resolves phoneme ambiguity, and

Table 4: The role of viseme conversion strategy and pseudo-audio generation module on GRID dataset.

| Methods | Semantic Quality | | | Video Quality | | | | |
|---|---|---|---|---|---|---|---|---|
| | BLEU-1↑ | BLEU-4↑ | WER↓ | SSIM↑ | PSNR↑ | LPIPS↓ | FID↓ | FVD↓ |
| Text | 40.04 | 9.91 | 52.64 | 0.681 | 15.896 | 0.301 | 187.628 | 3912.511 |
| Text → Viseme | 47.23 | 14.78 | 45.76 | 0.702 | 17.553 | 0.292 | 120.391 | 2673.519 |
| Text2Lip (Ours) | **54.81** | **23.50** | **39.43** | **0.740** | **19.023** | **0.238** | **32.109** | **762.683** |

significantly improves semantic quality indicators (BLEU-1 increases by 17.9% to 47.23, and WER decreases by 13.1% to 45.76). It also enhances lip movement coherence and promotes video quality optimization (SSIM increases by 0.702 and FID decreases to 120.391). Furthermore, the PVAR strategy gradually replaces audio features with text embeddings during training through curriculum learning, and collaborates with the pseudo-audio generation module to re-encode the audio features. The new method uses acoustic constraints to achieve a performance leap in the absence of audio: at the semantic level, BLEU-4 significantly increases by 59.1% to 23.50, and WER further decreases by 13.8% to 39.43; at the visual fidelity level, LPIPS increases by 18.5% to 0.238, PSNR increases by 8.4% to 19.023, and FVD decreases to 762.683, confirming the reasonable guidance of pseudo-audio on lip movements and verifying that Text2Lip provides an explainable technical paradigm for cross-modal generation.

**Influence of Different Audio Sources.** As shown in Table 5, we compare three audio sources: audio generated by IndexTTS, offline coding audio, and our proposed pseudo-audio. IndexTTS serves as a representative baseline model in this technical pathway. Our method achieves superior semantic qual-

Table 5: Comparison results of different audio sources on GRID.

| Sources | Semantic Quality | | | Video Quality | | | |
|---|---|---|---|---|---|---|---|
| | BLEU-1↑ | BLEU-4↑ | WER↓ | SSIM↑ | PSNR↑ | LPIPS↓ | FID↓ |
| GT | 55.20 | 24.45 | 39.48 | 0.743 | 19.217 | 0.223 | 30.296 |
| From IndexTTS | 49.46 | 16.82 | 44.29 | 0.718 | 17.631 | 0.281 | 91.276 |
| From offline coding | 51.86 | 19.72 | 41.84 | 0.721 | 17.859 | 0.276 | 87.655 |
| From viseme (Ours) | **54.81** | **23.50** | **39.43** | **0.740** | **19.023** | **0.238** | **32.109** |

ity (BLEU-1: 54.81, BLEU-4: 23.50, WER: 39.43) compared to both IndexTTS and offline encoding. In terms of visual quality, our method also surpasses the baselines in 0.740 on SSIM, 19.023 on PSNR and 32.109 on FID, demonstrating improvements in both semantic accuracy and visual real-

ism. Notably, our pseudo-audio achieves performance close to that of ground-truth audio, demonstrating its ability to effectively capture the speech features crucial for high-quality speaking face generation and effectively achieving the desired effect of audio replacement. Further validation of the advantage of our proposed end-to-end approach over traditional separated solutions under the same "audio-unavailable" constraints.

**Effect of Progressive Viseme-Audio Replacement.** In this part, we evaluate the fixed-probability random dropout strategies, which is a static training approach where all training steps face the same difficulty level, our strategy

Table 6: Comparison results of Progressive Viseme-Audio Replacement on GRID.

| Methods | BLEU-1↑ | BLEU-4↑ | WER↓ | SSIM↑ | PSNR↑ | LPIPS↓ | FID↓ |
|---|---|---|---|---|---|---|---|
| drop=30% | 48.72 | 17.62 | 45.11 | 0.713 | 17.916 | 0.288 | 93.764 |
| Text2Lip (Ours) | **54.81** | **23.50** | **39.43** | **0.740** | **19.023** | **0.238** | **32.109** |

establishes a learning path that better aligns with cognitive principles through a progressive easy-to-hard transition - gradually moving from abundant audio cues to complete audio absence. As shown in Table 6, Our curriculum learning approach significantly outperforms them across key metrics, including semantic quality, visual fidelity, and synchronization. This empirical validation confirms that this design enables the model to smoothly transition from audio dependency to text dependency, ultimately achieving stable convergence to the ideal "zero-audio" inference state.

## 4.4 QUALITATIVE RESULTS

**Performance on Unseen Speakers.** As shown in the visualization in Figure 4, we conducted validation analysis on unseen speakers in an open-domain scenario. For character settings with differentiated basic appearance features, such as boys with glasses and long-haired girls, the generated video sequences accurately reproduced the speech-synchronized lip movements and natural facial dynamics of the reference image while preserving key features of the target identity. Furthermore, in complex, blurred scenes, the algorithm effectively separated foreground characters from interfering background in-

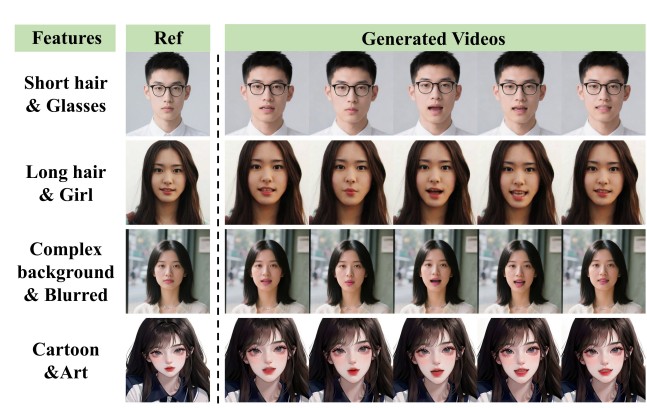

Figure 4: Visualization examples of unseen speakers in open domains.

formation. The generated dynamic images preserved the subject's facial details while also appropriately filling in the blurred areas through adaptive background generation, verifying the model's stability under the influence of environmental variables. In the challenging cartoon art style task, the generated results successfully captured the line features and color semantics of the hand-drawn style, transforming the real-life reference into a virtual avatar with a distinctive artistic style. The overall results show that the identity-separated lip-sync video generation strategy can be flexibly transferred to different roles and scenarios, providing an effective solution for speaker-unseen talking face generation in open domains.

**Performance on Unseen Sentences.** To verify its performance on unseen open-domain sentences, we use words from the training vocabulary to form novel, previously unseen sentence "please set eight red bins with green in one place" for visualization analysis. This is a standard and core paradigm for assessing compositional generalization Because the model's task is not to memorize specific sentences but to learn a generalizable mapping from text to facial motion. When presented with a new sentence composed of known words in a novel syntactic and semantic sequence, the model must dynamically interpret its meaning and correctly compose and sequence the learned phoneme-to-viseme mappings to generate coherent and accurate lip movements.

As shown in Figure 5, compared to mainstream methods such as Sonic, Hallo2, AniPortrait and V-Expres, our method significantly outperforms in lip shape naturalness, lip fluency, and semantic relevance: The lip shape space closely matches the articulatory movements (*e.g.*, lips open in "eight" and lips closed in "set"), and generalization is stable in the open domain (even robust to untrained sentences). The enlarged image further verifies that key phonemes (*e.g.*, lips open in the vowel "eight" /eɪ/ and lips and teeth contact in the final /t/) are strictly mapped to viseme lip shape, demonstrating the advantages of viseme decoupled representation. This method enables precise control of lip-synced talking faces generation in complex open-domain scenarios, providing technical support for cross-modal visual interaction.

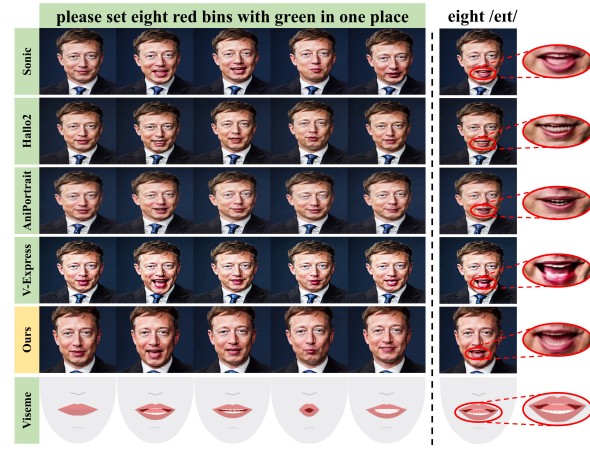

Figure 5: Visualization examples of unseen sentences in open domains.

**Visualization Results on GRID dataset.** Figure 6 shows a qualitative comparison between baseline methods (Sonic, Hallo2, AniPortrait and V-Express) and our method on GRID dataset, focusing on the third column (blink dynamics) and the sixth column (lip shape details) with significant differences. In terms of tone-driven blinks, the compared methods suffer from insufficient closure (*e.g.*, residual eyelid occlusion in Sonic and Hallo2) and abrupt transitions (*e.g.*, uniform occlusion lacking gradualness in AniPortrait). Our method accurately captures eyelid edge alignment and light-shadow transitions, and its closure duration and opening and closing amplitude are highly consistent with the GT. Regarding lip shape details, the comparisons exhibit blurred contours (*e.g.*, jagged edges in V-Express) and

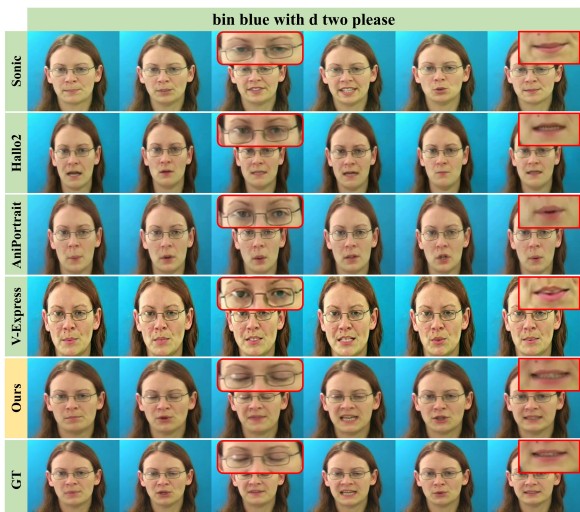

Figure 6: Visualization examples on GRID dataset.

motion asymmetry (*e.g.*, lip deviation in AniPortrait). Our method generates lip shapes with clear anatomical structure, a natural transition between the upper lip peak and lower lip valley that aligns with articulatory dynamics, and a closer approximation to the GT. The results show that Text2Lip can achieve fine coordination of eyebrow movement, eyelid opening and closing, and lip shape while keeping the head stable, and is overall closer to the natural human expressions.

## 5 CONCLUSIONS

In this work, we propose a viseme-centric framework, Text2Lip, that bridges linguistic semantics and visual articulation for viseme-driven lip-synced talking face generation. The proposed method addresses key limitations of conventional audio-driven approaches through three core innovations: structured viseme mapping for semantic-aligned lip motion priors, curriculum-based viseme-audio replacement for robust multi-modal handling, and landmark-guided rendering for photorealistic synthesis. Extensive evaluations demonstrate superior performance in both semantic consistency, visual quality, and generalization in open domains.

## ETHICS STATEMENT

This work does not involve human subjects, animal experiments, or sensitive personal data. The datasets used (*e.g.*, GRID, AVDigits) are publicly available benchmark datasets commonly used in machine learning research and do not contain personally identifiable information. We have carefully reviewed the ICLR Code of Ethics and confirm that this submission complies with its principles regarding fairness, privacy, and research integrity. No potential conflicts of interest exist among the authors.

## REPRODUCIBILITY STATEMENT

To ensure reproducibility, we provide the following resources: (1) All implementation details, including network architectures, hyperparameters, and training protocols, are described in Section 4.1 and the Appendix. (2) Random seeds are fixed, all results are averaged over multiple runs.

## LLM USAGE STATEMENT

Large Language Models (LLMs) were used in this work solely as a general-purpose writing assistance tool-for example, to improve grammar, clarify phrasing, or check technical terminology in the manuscript. LLMs did not contribute to the conception of the research idea. theoretical analysis, experimental design, or interpretation of results. All scientific content, including equations algorithms, and claims, was developed and verified by the authors. No LLM was used to generate novel technical content or to draft substantial portions of the paper. As required by ICLR policy, we confirm that LLMs are not listed as authors, and we take full responsibility for all content under our names.

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

# A  APPENDIX

## A.1  MORE TECHNICAL DETAILS

**Viseme Encoder & Audio Encoder Details**    In the Viseme encoder, we adopted a stacked structure, which consists of $n$ encoder blocks with exactly the same structure. Each encoder block contains three parts: a Multi-Head Attention layer ($MHA$), two Normalization Layers ($NL$) and a Feed-Forward Layer ($FL$). As shown in Figure 7 (a). The calculation process of each block can be expressed as:

$$\tilde{v}_n = NL(FL(MHA(NL(v'_n)) + v'_{n-1})), \tag{12}$$

where $v'_{n-1}$ is the viseme feature of the previous moment.

Correspondingly, as shown in Figure 7 (b), in the audio encoder part, we also use $n$ encoder modules with the same structure, and its calculation process can be formalized as:

$$\tilde{a}_n = NL(FL(MHA(NL(a'_n)) + a'_{n-1})), \tag{13}$$

where $a'_{n-1}$ is the audio feature of the previous moment.

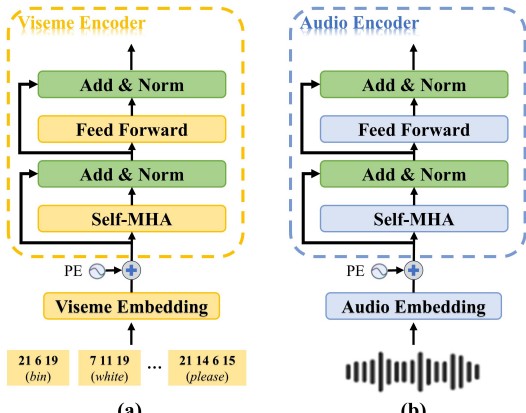

Figure 7: The details of Viseme Encoder & Audio Encoder.

## A.2  MORE EXPERIMENTAL DETAILS

### A.2.1  HYPER-PARAMETERS OF BASELINES

Table 7 presents the hyper-parameters of Text2Lip used in this work.

| Parameters | Viseme Encoder | Audio Encoder | Lip Decoder |
|---|---|---|---|
| layers | 2 | 2 | 2 |
| attention heads | 4 | 4 | 4 |
| hidden size | 512 | 512 | 512 |
| learning rate | $1 \times 10^{-3}$ | $1 \times 10^{-3}$ | $1 \times 10^{-3}$ |
| optimizer | Adam | Adam | Adam |
| dropout | 0 | 0 | 0 |
| batch-size | 128 | 128 | 128 |
| trg-size | – | 39 | 136 |

Table 7: Hyper-parameters of text-driven pose modalities co-generation.

### A.2.2  MORE DETAILS OF USER STUDY

Figure 8 shows a portion of the questionnaire. A total of 100 participants rated the performance of the five compared methods on a 5-point scale (1 to 5), with 1 representing "very unnatural/very out of sync" and 5 representing "very natural/perfectly synchronized." The results, summarized in Table 5, provide key insights into human perceptual preferences for different methods.

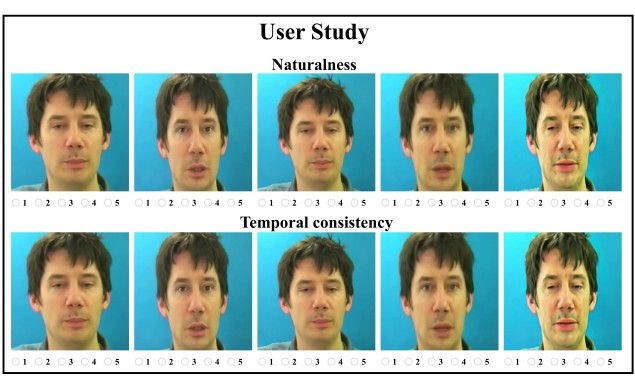

Figure 8: User study questionnaires.

### A.2.3 MORE DETAILS OF EVALUATION METRICS

Table 8: Evaluation indicator classification.

| Dimensions | Metrics |
| --- | --- |
| Visual similarity | SSIM, PSNR, LPIPS, FID |
| Landmark matching degree | DTW-P, MPJPE |
| Audio and lip consistency | Sync-C, Sync-D |
| Semantic quality | BLEU, WER |

In the experimental section, we use multiple metrics for quantitative analysis from different dimensions. First, we use **SSIM**, **PSNR**, **LPIPS** and **FID** to measure the visual similarity between the generated videos and the real videos at the image and video levels. **DTW-P** and **MPJPE** are used to measure the degree of match between the generated landmark sequences and the ground truth. Furthermore, we introduce the SyncNet model to compute **Sync-C** and **Sync-D**, where Sync-C measures the synchronization consistency between lip sounds and audio, while Sync-D evaluates the temporal consistency of dynamic lip movements. Furthermore, we design an inverse evaluation method to measure the semantic preservation accuracy of generated videos. Specifically, we retrain the NSLT model, which can interpret text sentences from lip-reading videos and compute text alignment accuracy, such as **BLEU** and **WER**. Subsequently, we will interpret the above evaluation metrics in more detail.

**SSIM (Structural Similarity Index):** The SSIM Wang et al. (2004) is a perceptual metric that quantifies image quality degradation by comparing structural information, luminance, and contrast between a reference and a distorted image. It ranges from -1 to 1, where 1 indicates perfect similarity.

- $x$: reference image patch
- $y$: distorted image patch
- $\mu_x, \mu_y$: mean intensities of $x$ and $y$
- $\sigma_x^2, \sigma_y^2$: variances of $x$ and $y$
- $\sigma_{xy}$: covariance between $x$ and $y$
- $C_1 = (K_1 L)^2, C_2 = (K_2 L)^2$: stabilization constants, where $L$ is the dynamic range (255 for 8-bit images), $K_1 = 0.01$, $K_2 = 0.03$
- The SSIM formula is:

$$\text{SSIM}(x,y) = \frac{(2\mu_x\mu_y + C_1)(2\sigma_{xy} + C_2)}{(\mu_x^2 + \mu_y^2 + C_1)(\sigma_x^2 + \sigma_y^2 + C_2)} \tag{14}$$

- For full images, Mean SSIM (MSSIM) is calculated by averaging SSIM values over all patches

**PSNR (Peak Signal-to-Noise Ratio):** The PSNR Hore & Ziou (2010) is a widely used metric for measuring the quality of reconstructed images. It quantifies the ratio between the maximum possible power of a signal and the power of corrupting noise, expressed in decibels (dB). Higher PSNR values indicate better image quality.

- $I$: reference image of size $m \times n$

- $K$: reconstructed image of same size

- $\text{MAX}_I$: maximum possible pixel value (255 for 8-bit images)

- MSE: mean squared error between images

$$\text{MSE} = \frac{1}{mn} \sum_{i=0}^{m-1} \sum_{j=0}^{n-1} [I(i,j) - K(i,j)]^2 \tag{15}$$

- The PSNR formula is:

$$\text{PSNR} = 10 \cdot \log_{10} \left( \frac{\text{MAX}_I^2}{\text{MSE}} \right) \tag{16}$$

**LPIPS (Learned Perceptual Image Patch Similarity):** The LPIPS Zhang et al. (2018) is a deep learning-based metric that measures perceptual similarity between images. Unlike traditional metrics, LPIPS uses features from pre-trained CNNs to better align with human perception, where lower values indicate higher similarity (0 = identical).

- $x, x_0$: reference and distorted image patches

- $F^l$: features extracted from layer $l$ of a pre-trained CNN (e.g., VGG or AlexNet)

- $w_l$: learned channel-wise weights for layer $l$

- Normalized feature difference:

$$\hat{d}^l = \frac{F^l(x) - F^l(x_0)}{\|F^l(x)\|_2} \tag{17}$$

- The LPIPS distance is computed as:

$$\text{LPIPS}(x, x_0) = \sum_l \frac{1}{H_l W_l} \sum_{h,w} \|w_l \odot \hat{d}^l_{h,w}\|_2^2 \tag{18}$$

where $H_l, W_l$ are spatial dimensions at layer $l$, and $\odot$ denotes channel-wise multiplication.

**FID (Fréchet Inception Distance):** The FID Heusel et al. (2017) measures the similarity between generated and real image distributions using features from the Inception network. Lower FID scores indicate better quality and diversity of generated images (perfect match = 0).

- $p_r$: real image distribution

- $p_g$: generated image distribution

- $\mu_r, \mu_g$: mean features of real and generated images from Inception-v3 pool3 layer

- $\Sigma_r, \Sigma_g$: covariance matrices of real and generated features

- The FID formula is:

$$\text{FID} = \|\mu_r - \mu_g\|^2 + \text{Tr}\left( \Sigma_r + \Sigma_g - 2\left(\Sigma_r \Sigma_g\right)^{1/2} \right) \tag{19}$$

where Tr denotes the matrix trace operation.

**DTW-P (Dynamic Time Warping Precision):** The DTW-P Sakoe & Chiba (2003) measures the alignment precision between two temporal sequences using dynamic programming. It evaluates how well a generated sequence matches a reference sequence in terms of temporal dynamics, with lower values indicating better alignment (perfect match = 0).

- $X = [x_1, x_2, ..., x_m]$: reference sequence of length $m$
- $Y = [y_1, y_2, ..., y_n]$: generated sequence of length $n$
- $d(x_i, y_j)$: local distance between elements $x_i$ and $y_j$
- The DTW distance is computed as:

$$D(i,j) = d(x_i, y_j) + \min \begin{cases} D(i-1, j) \\ D(i, j-1) \\ D(i-1, j-1) \end{cases} \tag{20}$$

- DTW-P is the normalized alignment error:

$$\text{DTW-P} = \frac{D(m,n)}{m+n} \tag{21}$$

**MPJPE (Mean Per Joint Position Error):** The MPJPE Ionescu et al. (2013) is a standard metric for evaluating 3D human pose estimation accuracy. It measures the average Euclidean distance between predicted and ground truth joint positions, with lower values indicating better pose reconstruction (perfect match = 0 mm).

- $J$: set of human joints (e.g., 17 joints for COCO format)
- $\hat{\mathbf{p}}_j$: predicted 3D position of joint $j$
- $\mathbf{p}_j$: ground truth 3D position of joint $j$
- The MPJPE formula is:

$$\text{MPJPE} = \frac{1}{|J|} \sum_{j \in J} \|\hat{\mathbf{p}}_j - \mathbf{p}_j\|_2 \tag{22}$$

where $\|\cdot\|_2$ is the Euclidean norm.

**Sync-C (Synchronization confidence):** The Sync-C Chung & Zisserman (2016) measures the audio-visual synchronization confidence between speech and lip movements. Higher values (0-5 scale) indicate better synchronization quality.

- $a_t$: audio feature vector at time $t$
- $v_t$: visual feature vector at time $t$
- The synchronization confidence is computed as:

$$\text{Sync-C} = 5 \cdot \sigma \left( \frac{1}{T} \sum_{t=1}^{T} \text{cosine}(a_t, v_t) \right) \tag{23}$$

where $\sigma$ is the sigmoid function, and $T$ is the number of frames.

**Sync-D (Synchronization Distance):** The Sync-D Chung & Zisserman (2016) quantifies the misalignment between audio and visual streams as the average feature distance. Lower values indicate better synchronization.

- $\mathbf{a}, \mathbf{v}$: SyncNet embeddings for audio and video segments
- The synchronization distance is:

$$\text{Sync-D} = \frac{1}{N} \sum_{i=1}^{N} \|\mathbf{a}_i - \mathbf{v}_i\|_2 \tag{24}$$

where $N$ is the number of segment pairs.

**BLEU (Bilingual Evaluation Understudy):** The BLUE Papineni et al. (2002) is a commonly used metric to assess the quality of machine translation. It is calculated by the following formula:

$$\text{BLEU} = \text{BP} \cdot \exp\left(\sum_{n=1}^{N} w_n \cdot \log p_n\right) \tag{25}$$

where:

- BP (Brevity Penalty): A brevity penalty factor used to penalize candidate translations that are shorter than the reference translations.

$$\text{BP} = \begin{cases} 1 & \text{if } c > r \\ e^{(1-\frac{r}{c})} & \text{if } c \leq r \end{cases} \tag{26}$$

- $c$ denotes the length of the candidate translation.
- $r$ denotes the length of the reference translation.
- $p_n$: The precision for n-grams, defined as the number of matching n-grams between the candidate and reference translations divided by the total number of n-grams in the candidate translation.

$$p_n = \frac{\text{Number of matched n-grams}}{\text{Total number of n-grams in candidate translation}} \tag{27}$$

- $w_n$: The weight assigned to each n-gram precision, typically set as $w_n = \frac{1}{N}$, where $N$ represents the maximum length of the n-grams considered (commonly $N = 4$, covering 1-gram through 4-gram precision).
- $exp$: Represents the exponentiation of the sum of the weighted logarithmic precisions.

In our main experiments, we mainly use BLEU-1 and BLEU-4 scores to reflect the accuracy of word-level translation and the quality of overall sentence translation, respectively.

**WER (Word Error Rate):** The WER Assael et al. (2016) is a commonly used evaluation metric to measure the accuracy of a translation system. WER measures the error rate in the generated text, accounting for three types of errors: substitutions, insertions, and deletions. It is expressed as the ratio of the total number of errors to the total number of words in the reference text. The formula for calculating WER is:

$$\text{WER} = \frac{S + D + I}{N} \tag{28}$$

where:

- $S$ represents the number of substitutions.
- $D$ represents the number of deletions.
- $I$ represents the number of insertions.
- $N$ is the total number of words in the reference text.

