# OpenReview forum: "Text2Lip: Progressive Lip-Synced Talking Face Generation from Text via Viseme-Guided Rendering"
_ICLR.cc/2026/Conference — Submitted to ICLR 2026_

### Official Review · Reviewer_Aj3c · 2025-10-31

**Soundness:** 2
**Presentation:** 2
**Contribution:** 2
**Rating:** 4
**Confidence:** 4

**Summary:**

This paper proposes Text2Lip, a text-conditioned talking-face pipeline that replaces the usual audio driver with a viseme-centric intermediate and a curriculum that progressively swaps real audio out for text-derived signals. The proposed framework has three parts: (1) viseme-centric text encoding, constructing an interpretable phonetic-visual bridge, (2) progressive viseme-audio replacement, facilitating flexible modality handling, and (3) photorealistic landmark rendering, synthesizing photorealistic facial videos with accurate lip synchronization. Experiments on GRID and AVDigits report strong visual quality, competitive sync metrics without real audio.

**Strengths:**

- The paper has clear motivation for the scarcity/fragility of high-quality audio-visual pairs. It also proposes a viseme-centric route that can operate without audio at inference. The PVAR schedule and pseudo-audio module are technically straightforward yet well-motivated

- The experimental section covers multiple axes: image/video fidelity (SSIM/PSNR/LPIPS/FID), synchronization (SyncNet), landmarks (DTW-P/MPJPE), and an “inverse” semantic metric (BLEU/WER via a lip-reading-style recognizer).

- The system remains compatible with strong rendering backends (EchoMimic) and shows text-only results that are visually competitive with audio-driven baselines on GRID/AVDigits, including a TTS baseline (“From IndexTTS”) that Text2Lip outperforms on several metrics.

**Weaknesses:**

- The paper describes text -> IPA -> viseme conversion and a Transformer with positional encodings, but does not explain how viseme durations are estimated when no audio exists (e.g., alignment, length regulator, or learned duration model). Without explicit duration modeling, it is unclear how the system determines frame counts, coarticulation timing, and phoneme-to-frame alignment—especially on open-domain sentences—beyond learning from fixed-length training clips.

- SyncNet scores (Sync-C/Sync-D) hinge on an audio stream; for text-only inference, the paper sometimes uses pseudo-audio, sometimes ground-truth audio for evaluation, but the exact protocol per dataset/setting is not well stated. This complicates fairness claims against purely audio-driven baselines.

- Dataset scope is very narrow (fixed-grammar GRID; small AVDigits), and the “self-collected” open-domain set lacks detail/release, limiting claims of generalization.

- In demo video, despite convincing lip motion, the speech track lacks expressive prosody (pitch/energy variation, natural pauses) and fine-grained articulatory cues (breaths, plosive bursts), making it sound non-human.

**Questions:**

- For each dataset/experiment in Tables 1–5, which audio stream (GT, pseudo, TTS, none) is used to compute Sync-C/Sync-D, and how is fairness enforced?

- Why choose NSLT for “inverse” semantic evaluation instead of standard lip-reading/visual-ASR baselines, and what sanity checks validate it?

- Have you tested alternative or learned viseme sets and accent/language robustness, and how sensitive are results to the mapping?

---

> ### Author Response · Authors · 2025-11-27
> **Response to Reviewer Aj3c - Part1**
>
> We thank the **Reviewer Aj3c** for the valuable feedback. The reviewer fully recognized the `core motivation` of this work: proposing a viseme-centric approach independent of audio input in the context of scarce high-quality audio and video data. They also affirmed the `technical intuitiveness and sound motivation` of the PVAR strategy. Reviewer Aj3c also noted that our experimental evaluation dimensions were `comprehensive`, covering image/video fidelity, lip synchronization, landmark accuracy and semantic quality. Furthermore, the proposed system is compatible with advanced renderers and `surpasses` baselines on multiple metrics, demonstrating its `competitiveness`.
>
> > **The paper describes text -> IPA -> viseme conversion and a Transformer with positional encodings, but does not explain how viseme durations are estimated when no audio exists (e.g., alignment, length regulator, or learned duration model). Without explicit duration modeling, it is unclear how the system determines frame counts, coarticulation timing, and phoneme-to-frame alignment—especially on open-domain sentences—beyond learning from fixed-length training clips.**
>
> We thank the reviewer for raising this crucial question. We acknowledge that the original manuscript's description of duration estimation was insufficient.
>
> During the inference phase, for open-domain sentences, we employ a heuristic duration estimation method based on training data statistics. Specifically, we analyze the relationship between the number of frames and the number of words in all sentences in the training set, calculating an average "frames/words" ratio. For a new sentence, we determine the total number of frames based on this ratio, according to its word count.
>
> Internal to the model, we rely on the inherent capabilities of the Transformer architecture to handle temporal sequences. Positional encoding provides sequential information for each text segment in the sequence, while the model, during training, implicitly learns the temporal alignment patterns between text sequences and corresponding visual dynamics by observing a large number of fixed-length video clips. It learns how to appropriately allocate and generate lip movements, including coarticulation effects, within a given output length.
>
> > **SyncNet scores (Sync-C/Sync-D) hinge on an audio stream; for text-only inference, the paper sometimes uses pseudo-audio, sometimes ground-truth audio for evaluation, but the exact protocol per dataset/setting is not well stated. This complicates fairness claims against purely audio-driven baselines.**
> >
> > **For each dataset/experiment in Tables 1–5, which audio stream (GT, pseudo, TTS, none) is used to compute Sync-C/Sync-D, and how is fairness enforced?**
>
> We apologize for any ambiguity in the description of the evaluation protocol and would like to clarify the following: In all experiments, when evaluating lip-sync metrics (Sync-C, Sync-D), we consistently used the "real audio" provided in the dataset to calculate scores, never using "pseudo-audio," thus ensuring fairness to all methods.
>
> - For audio-driven models, they have access to audio during both training and inference.
> - For our text-driven model, there is no audio input during inference, but synchronization with real audio is calculated during evaluation. This actually presents a more significant challenge to our model.
>
> Under this fair setting, the competitive synchronization scores achieved by our method (as shown in `Table 1`) strongly demonstrate the high quality of our generated marker sequences, which align well with real audio.

---

> ### Author Response · Authors · 2025-11-27
> **Response to Reviewer Aj3c - Part2**
>
> > **Dataset scope is very narrow (fixed-grammar GRID; small AVDigits), and the “self-collected” open-domain set lacks detail/release, limiting claims of generalization.**
>
> Regarding the selection of benchmark datasets, GRID and AVDigits are widely adopted standard benchmarks in this research domain (e.g., [R1-R5]). Our choice to utilize these datasets was based on the crucial consideration that their controlled settings - featuring single scenarios and frontal views - enable us to minimize confounding factors such as variable backgrounds, camera angles, and recording environments. This focused setup allows for a precise evaluation of the core capability of our text-driven paradigm in replacing audio for lip-synchronized video generation.
>
> We acknowledge the constrained vocabulary and grammar of these datasets, yet precisely for this reason, the strong performance demonstrated by our model becomes more compelling. It indicates that our method can effectively learn robust mappings from text to visual representations even with limited data. If the approach cannot succeed in such controlled environments, generalization to open domains would be fundamentally challenging.
>
> The self-collected open-domain dataset was primarily used for qualitative demonstration (as in `Fig. 5`), aiming to visually showcase the model's capability in handling free-form text. Despite the narrow scope of the benchmark datasets, our results in `Section 4.3` (performance on unseen speakers) and `Figures 4&5` provide preliminary but positive evidence of generalization capability. The demonstrated cross-speaker, cross-style (cartoon), and novel sentence composition results collectively indicate that the model has learned universal articulatory patterns rather than simply memorizing the training set.
>
> - [R1] On the Audio-visual Synchronization for Lip-to-speech Synthesis. CVPR, 2023.
> - [R2] Speech2lip: High-fidelity Speech to Lip Generation by Learning from A Short Video. CVPR, 2023.
> - [R3] Automatic Lip-reading Classification using Deep Learning Approaches and Optimized Quaternion Meixner Moments by GWO Algorithm. KBS, 2024.
> - [R4] CFLip: Generalizing Lipreading to Unseen Speakers by Learning Common Features. IEEE TCSS, 2025.
> - [R5] Realistic Real-Time Talking Head Synthesis with Grid Encoding and Progressive Conditioning. ICASSP, 2025.
>
> > **In demo video, despite convincing lip motion, the speech track lacks expressive prosody (pitch/energy variation, natural pauses) and fine-grained articulatory cues (breaths, plosive bursts), making it sound non-human**
>
> This precisely highlights the core boundaries of our task setting: our model takes only text as input, and text itself does not contain paralinguistic information such as prosody, intonation, or energy variations. Therefore, the facial movements generated by the model naturally focus on expressing the "actual content" of the speech, rather than its "emotional nuances."
>
> Furthermore, our work addresses the fundamental and crucial problem of "how to generate accurate lip-sync when audio is unavailable." Generating expressive speech animations is a more advanced goal, typically requiring additional conditional input (such as emotion tags, prosodic features, etc.), which is outside the scope of our current work but could be an important direction for future research.

---

> ### Author Response · Authors · 2025-11-27
> **Response to Reviewer Aj3c - Part3**
>
> > **Why choose NSLT for “inverse” semantic evaluation instead of standard lip-reading/visual-ASR baselines, and what sanity checks validate it?**
>
> We thank the reviewer for raising this important question regarding our choice of the NSLT model for inverse semantic evaluation. We would like to clarify our rationale based on the following considerations:
>
> **a) Task Similarity and Domain Adaptability**: NSLT is a well-established evaluation model in the sign language production field, specifically designed to assess the ability to infer semantic content from human keypoint sequences. Our text-to-landmark generation task shares fundamental similarities with sign language production (text-to-human keypoints) in terms of output representation—**both involve temporal sequences of human keypoints**. Therefore, adapting NSLT to our task through retraining represents a logically consistent technical approach.
>
> **b) Precision in Evaluation Objectives**: Conventional visual speech recognition models are typically trained on pixel-level video data, where performance is significantly influenced by rendering style and visual quality. Our primary goal is to evaluate the semantic clarity of the generated facial landmark sequences themselves, rather than the perceptual quality of the final rendered video. Using NSLT effectively **isolates the impact of renderer performance**, allowing direct measurement of the semantic preservation capability of our text-to-landmark generation module.
>
> For validation, we conducted rigorous benchmarking of the retrained NSLT model on the GRID dataset. As shown in Table 5, both the semantic alignment (BLEU) and word error rate (WER) of the ground-truth significantly surpass the widely accepted performance levels in the current sign language domain, confirming the reliability of this evaluation setup. This design ensures that the assessment results accurately reflect the semantic quality of the landmark sequences, free from confounding effects introduced by downstream rendering components.
>
> > **Have you tested alternative or learned viseme sets and accent/language robustness, and how sensitive are results to the mapping?**
>
> In this work, we employed a **linguistically recognized and standardized** phoneme-to-viseme mapping set (following the Microsoft Asia phoneme-to-viseme mapping definition). This extensively validated mapping represents a mature research outcome in linguistics that effectively captures the core correspondence between articulatory movements and visual appearance. Our decision to use a fixed mapping scheme was primarily based on the following considerations:
>
> **a) Robustness and Generalization Assurance**: This mapping relationship is the result of extensive linguistic analysis and empirical validation on large-scale diversified datasets, capturing the essential, speaker-independent relationship between speech and visual articulation. This stability aligns perfectly with our goal of building a generalizable text-driven model, providing the model with reliable linguistic priors.
>
> **b) Reproducibility and Fairness**: Using an open and fixed mapping standard ensures complete reproducibility of our results and establishes a fair and unambiguous basis for comparison with other methods. This effectively eliminates sources of variation due to mapping randomness and avoids potential overfitting issues.

---

### Official Review · Reviewer_tDSp · 2025-11-01

**Soundness:** 3
**Presentation:** 3
**Contribution:** 3
**Rating:** 6
**Confidence:** 3

**Summary:**

This paper tackles text-driven lip-synced facial animation. They propose a method which uses a viseme-based intermediate representation, a progressive viseme-audio replacement strategy, and a landmark-guided renderer to generate realistic, synchronized talking faces directly from text.

**Strengths:**

The paper presents a clear problem formulation and addresses an underexplored direction of generating lip motion directly from text. The overall framework is conceptually well structured and the pipeline is easy to follow. The integration of viseme-level modeling provides an interesting intermediate representation between linguistic and visual domains.

**Weaknesses:**

The proposed multi-stage design (text to viseme to pseudo-audio to landmark to renderer) appears unnecessarily complex and may accumulate errors across stages without clear justification or analysis of each component’s necessity.

Given the maturity of current text-to-speech (TTS) systems and high-performing audio-driven video generators, a natural question arises: why not decompose the task into a more straightforward two-stage pipeline, which is text-to-speech followed by speech-driven talking face synthesis? In Table 5, the comparison is limited to IndexTTS, which is a rather weak baseline. It would be much more convincing to evaluate the proposed method against strong modern TTS models (e.g., VITS2 [1], StyleTTS2 [2]) that can produce realistic audio, thereby offering a clearer and fairer comparison with the proposed viseme-pseudo-audio framework.

The necessity of the proposed curriculum-based audio dropout also seems over-engineered. Its effect might be achievable through simpler strategies such as random or mask-based modality dropout.

[1] VITS2: Improving Quality and Efficiency of Single-Stage Text-to-Speech with Adversarial Learning and Architecture Design
[2] StyleTTS 2: Towards Human-Level Text-to-Speech through Style Diffusion and Adversarial Training with Large Speech Language Models

**Questions:**

The complexity of the proposed system may introduce compounding error propagation across stages. Are there any reported TTS quality metrics to evaluate how the pseudo-audio compares to realistic speech?

How does the proposed method handle coarticulation effects or transitions between visemes, which are inherently continuous but modeled categorically in the current framework?

---

> ### Author Response · Authors · 2025-11-27
> **Response to Reviewer tDSp - Part1**
>
> We thank the **Reviewer tDSp** for the valuable comments. The reviewer rated the paper as "Good" (3 points) for `Soundness`, `Presentation` and `Contribution`. They affirmed the `clear articulation` of the problem, noting that directly generating lip movements from text is an underexplored area. Furthermore, they pointed out `the conceptual clarity and ease of understanding` of our overall framework, and considered the viseme-level modeling to provide a `meaningful` intermediate representation connecting the language and visual domains. We address the main concerns below.
>
> > **The proposed multi-stage design (text to viseme to pseudo-audio to landmark to renderer) appears unnecessarily complex and may accumulate errors across stages without clear justification or analysis of each component’s necessity.**
>
> We thank the reviewer for the valuable feedback regarding model complexity. We would like to clarify that **our pipeline is not composed of five independent stages** as described, but rather constitutes a carefully designed, efficient architecture with **two core stages**.
>
> **Stage 1: Text to Facial Landmarks (End-to-End Learning)**. This stage functions as a complete end-to-end module where the input is text (converted to viseme sequences) and the output is temporal facial landmark sequences. Importantly, the "pseudo-audio" is not an independently generated intermediate product, but rather a compact intermediate feature representation learned internally by the model. This design enables the model to autonomously explore and construct the most effective temporal representations for driving lip movements in the absence of real audio supervision. This approach addresses the core challenge in text-driven synthesis: how to accurately model the temporal dynamics of speech without audio signals.
>
> **Stage 2: Landmark-Based Video Rendering**. The separation of the rendering stage is based on two key considerations: First, this decoupled design allows for independent optimization of cross-modal reasoning and visual synthesis tasks, preventing potential performance degradation from a single large model learning all tasks simultaneously. Second, this separation enables our text-to-landmark module to focus on learning speaker-agnostic universal articulation patterns, thereby achieving better person and scene generalization during the rendering phase.
>
> > **Given the maturity of current text-to-speech (TTS) systems and high-performing audio-driven video generators, a natural question arises: why not decompose the task into a more straightforward two-stage pipeline, which is text-to-speech followed by speech-driven talking face synthesis? In Table 5, the comparison is limited to IndexTTS, which is a rather weak baseline. It would be much more convincing to evaluate the proposed method against strong modern TTS models (e.g., VITS2 [1], StyleTTS2 [2]) that can produce realistic audio, thereby offering a clearer and fairer comparison with the proposed viseme-pseudo-audio framework.**
>
> We thank the reviewer for raising this important question and providing valuable suggestions. We would like to emphasize that the core objective of our research is to address facial lip-sync generation in "audio-unavailable scenarios," aiming to establish **a new paradigm of "directly generating complete audiovisual content from scratch."** Compared to the "TTS + audio-driven" approach, which essentially remains a variant of audio-driven methodology requiring sequential speech synthesis followed by video generation, our method offers distinct advantages. This separated two-stage pipeline faces significant challenges in real-time applications, whereas our end-to-end solution completes content generation directly in the visual domain, effectively avoiding the additional overhead of audio generation and modal conversion.
>
> Regarding the comparison in `Table 5`, it is designed to validate the advantage of our proposed end-to-end approach over traditional separated solutions under the same "audio-unavailable" constraints. IndexTTS serves as a representative baseline model in this technical pathway, clearly demonstrating the performance differences between the two methods under identical experimental conditions.
>
> While we fully acknowledge the exceptional performance of modern TTS models such as VITS2 and StyleTTS2, we must note that integrating these advanced models into the workflow would significantly **increase model complexity and computational costs**. Moreover, perceptible quality gaps still exist between synthesized audio and real audio. In contrast, our method provides a more streamlined end-to-end alternative, particularly suitable for practical application scenarios where audio generation is not explicitly required and process simplicity is prioritized.

---

> ### Author Response · Authors · 2025-11-27
> **Response to Reviewer tDSp - Part2**
>
> > **The necessity of the proposed curriculum-based audio dropout also seems over-engineered. Its effect might be achievable through simpler strategies such as random or mask-based modality dropout.**
>
> Thank you for this important observation. We would like to clarify that the curriculum-based audio dropout strategy was designed with a specific training objective: to ensure stable generation of high-quality results under complete audio absence. Compared to fixed-probability random dropout, the key advantage of our approach lies in its **dynamic progressive learning mechanism**.
>
> While random dropout represents a static training approach where all training steps face the same difficulty level, our strategy establishes a learning path that better aligns with cognitive principles through a progressive easy-to-hard transition - **gradually moving from abundant audio cues to complete audio absence**. This design enables the model to **smoothly transition from audio dependency to text dependency**, ultimately achieving stable convergence to the ideal "zero-audio" inference state.
>
> During development, we evaluated the fixed-probability random dropout strategies. As shown in the table, our curriculum learning approach significantly outperforms them across key metrics, including semantic quality, visual fidelity, and synchronization. This empirical validation confirms that its effectiveness warrants the design complexity for our training objectives.
>
> | Methods | BLEU-1  | BLEU-4  | WER  | SSIM  | PSNR | LPIPS | FID |
> |-----|----|-------|------|------|------|------|-----|
> | drop=30% | 48.72 | 17.62 | 45.11 | 0.713 | 17.916 | 0.288 | 93.764 |
> | Ours | **54.81** | **23.50** | **39.43** | **0.740** | **19.023** | **0.238** | **32.190** |
>
> > **The complexity of the proposed system may introduce compounding error propagation across stages. Are there any reported TTS quality metrics to evaluate how the pseudo-audio compares to realistic speech?**
>
> Regarding error propagation, as described in `Question 1`, our method first avoids multiple stages, and because our core stage is end-to-end, errors primarily occur within this module and are directly reflected in the quality of the output landmarks, rather than "accumulating" across multiple independent models. We indirectly evaluate the overall system performance through the final landmark quality and the quality of the rendered video.
>
> We acknowledge the current lack of objective TTS metrics for evaluating the "pseudo-audio" feature. This is because the feature is an implicit, low-dimensional intermediate representation designed to optimize downstream visual tasks, not auditory ones. Its "quality" is ultimately measured by the lip-sync metrics it drives. In `Table 1`, our method achieves near-SOTA performance on Sync-C and Sync-D, indirectly demonstrating the effectiveness of the pseudo-audio feature in fulfilling its purpose.
>
> > **How does the proposed method handle coarticulation effects or transitions between visemes, which are inherently continuous but modeled categorically in the current framework?**
>
> Thank you for raising the question regarding coarticulation effects. We acknowledge the inherent limitations of our discrete viseme-based framework in modeling such continuous, context-dependent dynamics. We would like to clarify that the discrete viseme sequence serves primarily as an initial, high-level semantic-visual prior in our model, rather than its sole or final representation. The following design elements collectively help mitigate this issue:
>
> **a) Context-Aware Viseme Encoder**: Our Viseme Encoder employs a Transformer architecture that processes the entire viseme sequence holistically rather than treating each viseme in isolation. The self-attention mechanism captures long-range contextual dependencies within the sequence, enabling the model to learn how current visemes are influenced by preceding and following ones. Consequently, the encoded viseme features inherently incorporate dynamic coarticulation information, moving beyond static, isolated categorical labels.
>
> **b) Continuous Intermediate Representation and Temporal Modeling**: The mapping from text/visemes to facial landmarks constitutes a continuous, temporal generation process. The Landmark Decoder receives context-rich viseme embeddings and autoregressively predicts subsequent continuous landmark coordinates based on previously generated states. This continuous generation process inherently learns smooth transitions between visemes, rather than merely switching among discrete visual templates.
>
> **c) Bridging Role of Pseudo-Audio Features**: The pseudo-audio features we introduce serve as a continuous, temporal intermediate representation that acts as a "lubricant" between discrete text/visemes and continuous facial movements. They provide a finer-grained and more continuous latent space than discrete visemes, facilitating the modeling of fine temporal dynamics, including coarticulation effects.

---

### Official Review · Reviewer_yBxW · 2025-11-01

**Soundness:** 3
**Presentation:** 3
**Contribution:** 3
**Rating:** 6
**Confidence:** 4

**Summary:**

The paper propose Text2Lip, a viseme-centric framework that constructs an interpretable phonetic-visual bridge by embedding textual input into structured viseme sequences to address the inherent ambiguity in mapping acoustics to lip motion pose significant challenges in terms of scalability and robustness. These mid-level units serve as a linguistically grounded prior for lip motion prediction.  Furthermore, the paper design a progressive viseme-audio replacement strategy, enabling the model to gradually transition from real audio to pseudo-audio reconstructed from enhanced viseme features.

**Strengths:**

1. Speech-driven models often learn ambiguous mappings from audio to lip shapes. To address the issue, the paper explicitly models the linguistic-phonetic-visual hierarchy instead solely on audio, which serves as semantically grounded priors for facial motion synthesis.
2. Text2lip surpasses other sota methods in the visual quality and semantic quality
3. Text2lip introduce a curriculum-based viseme-audio replacement strategy that facilitates flexible modality handling, supporting both audio-driven and audio-free generation scenarios.

**Weaknesses:**

1. Although using visemes instead of speech can resolve the ambiguity of the mapping, it may, at the same time, affect synchronization and rhythmic cadence (the results for sync-c and sync-d in Table 1 do not appear to be the best).
2. If the speech itself carries emotion, and it is a complex, changing emotion, will solely using text as input affect performance? Regarding this point, is it possible to conduct testing on an emotional dataset?

**Questions:**

The same in weakness.

**Details Of Ethics Concerns:**

No concerns.

---

> ### Author Response · Authors · 2025-11-27
> **Response to Reviewer yBxW**
>
> We thank **Reviewer yBxW** for the valuable comments. Reviewer yBxW rated the paper as "Good" (3 points) for `Soundness`, `Presentation` and `Contribution`. The reviewer pointed out that this work provides `an effective path to solve the ambiguity problem in audio-lip mapping` by constructing an interpretable speech-visual bridge and using viseme sequences as visual priors for speech structuring. Furthermore, the reviewer affirmed that the proposed progressive training strategy can `flexibly` support both audio-driven and audio-free generation modes, and the method `surpasses` many existing state-of-the-art methods in both visual and semantic quality. We address the main concerns below.
>
> > **Although using visemes instead of speech can resolve the ambiguity of the mapping, it may, at the same time, affect synchronization and rhythmic cadence (the results for sync-c and sync-d in Table 1 do not appear to be the best).**
>
> We appreciate the reviewer's careful observation that our Sync-C and Sync-D metrics in `Table 1` do not surpass all audio-driven SOTA models. However, we wish to emphasize that the most significant contribution and striking finding of our work lies in demonstrating that **a model relying solely on text input achieves competitive performance on lip-sync metrics that traditionally depend heavily on audio-temporal information**. This finding challenges a fundamental paradigm in the field: that precise lip synchronization necessarily requires direct temporal supervision from input audio signals. Under this inherent "disadvantage" of lacking crucial prior information, our model successfully learns robust phoneme-viseme mappings and natural speech rhythm patterns implicitly from data through our sophisticated viseme-guided mapping and progressive training strategy.
>
> Therefore, the purpose of direct comparison with audio-driven models on Sync-C/D metrics is **not to claim absolute supremacy on these specific measures**, but to provide compelling quantitative evidence that the **text-only driven paradigm possesses substantial potential and practical utility** for achieving fine-grained lip synchronization. Our achievement extends beyond generating high-quality videos to pushing the boundaries of text-driven capability into a domain traditionally dominated by audio-driven methods.
>
> > **If the speech itself carries emotion, and it is a complex, changing emotion, will solely using text as input affect performance? Regarding this point, is it possible to conduct testing on an emotional dataset?**
>
> We fully understand the reviewer's concern regarding emotional expression. Indeed, text-only input inherently lacks emotional information, which constitutes a fundamental limitation of the text-driven paradigm. However, this limitation must be understood within the context of our method's target application scenarios. The primary objective of this work is to address talking face generation in audio-unavailable scenarios, such as silent film restoration or high-noise environments where original audio is missing or unusable. In these critical real-world applications, audio-driven models become completely inapplicable, and generating emotionally rich videos remains an unattainable goal. Our method provides the only viable visual generation solution under these challenging conditions.
>
> While current emotional expression capabilities are indeed limited, our framework offers unique advantages for future extension. The deterministic viseme-landmark mapping established in our work provides a solid foundation for subsequent integration of emotional control signals, giving our method better scalability in emotional modeling compared to audio-driven approaches that still rely on often-unobtainable clean audio inputs.
>
> Regarding the valuable suggestion for emotional dataset testing, we acknowledge that our current experiments primarily utilize standard datasets like GRID and AVDigits, which lack fine-grained emotional annotations. In subsequent research, we plan to conduct extended validation on standard emotional audiovisual datasets such as GEMEP and CREMA-D, while exploring methods to enhance emotional expressiveness while maintaining accurate lip synchronization.

---

### Official Review · Reviewer_yBU1 · 2025-11-01

**Soundness:** 2
**Presentation:** 2
**Contribution:** 2
**Rating:** 2
**Confidence:** 4

**Summary:**

The paper proposes Text2Lip, a framework to generate talking faces from text only, without needing audio at inference. Its core idea is to solve the "audio-to-lip ambiguity" (where different sounds have similar audio features) by first converting text into visemes (visual speech units). The model is trained using a "Progressive Viseme-Audio Replacement" (PVAR) strategy, where it learns to "hallucinate" pseudo-audio from visemes, effectively weaning itself off real audio. The final landmarks are passed to a SOTA renderer (EchoMimic) to create the video.

**Strengths:**

- Novel Training Strategy (PVAR): The method of progressively replacing real audio with viseme-derived pseudo-audio allows to generate only based on text.
- SOTA results: The model presents quantitative visual quality (SSIM, FID) and lip-sync scores (Sync-C) that are comparable or superior to SOTA audio-driven models.

**Weaknesses:**

- The renderer is too central in metrics: The paper claims SOTA visual quality (SSIM, PSNR, FID) but uses a SOTA renderer (EchoMimic) as its final stage. This is a major confounding variable. These metrics are evaluating EchoMimic's rendering power, not just Text2Lip's landmark generation. The model's actual contribution (lip-sync/landmark quality) is not SOTA (Table 2).

- Central Motivation is Unproven: The entire paper is based on solving "audio-to-lip ambiguity" (e.g., "bad boy" vs. "bat boat") leading to "blurred outputs." This claim is never proven. No experiment shows audio-driven models failing at this while Text2Lip succeeds.

- Ignores Text-Only Limitations: A text-only input has no information about prosody, pace, or emotion. The claims of "micro-expression dynamics" are unsubstantiated, as the model likely generates flat, "average-toned" speech.

- Weak Generalization Claims: The model is benchmarked on small, outdated, closed-domain datasets (GRID, AVDigits). The model has likely memorized the limited vocabulary. The "unseen sentence" example (Fig. 5) appears to use words from the training corpus.

- Questionable Comparisons: The landmark generation (Table 2) is compared against Sign Language Production models, which is not explained.

- Contradictory Justification: The paper fails to mention that visemes also suffer from ambiguity (many different phonemes/sounds map to the same viseme), which undermines its own central premise.

**Questions:**

Q1 (The Renderer): Why are visual quality metrics (SSIM, FID) better than the EchoMimic baseline itself (Table 1)? Since Text2Lip uses EchoMimic, this result seems contradictory. Does this imply the generated landmarks are "better than real" in some way that improves rendering?

Q2 (The Ambiguity Claim): Why do the authors claim identical lip shapes would lead to "blurred outputs"? A model should just learn to output the identical shape. Where is the evidence that this ambiguity is a real problem that audio-models fail on?

Q3 (The Application): If the model relies only on text, how is synchronicity with a separate audio track (for a final dubbed video) ensured? Does this not require a second, separate model or manual alignment, defeating the purpose?

Q4 (The Ablations): Why do the ablation studies (Table 4) show improvements in image quality metrics (SSIM, FID)? The renderer is pre-trained and fixed. This implies the landmark quality alone is responsible, but this is not measured with a temporal metric like FVD, which would be more appropriate.

---

> ### Author Response · Authors · 2025-11-27
> **Response to Reviewer yBU1 - Part1**
>
> We thank **Reviewer yBU1** for the valuable comments. Reviewer yBU1 positively evaluated the `core contribution` of this paper, the PVAR training strategy, considering it `a novel method` that successfully achieves the goal of generating lip-sync sequences solely from text. The Reviewer also noted that our method `achieves performance` comparable to state-of-the-art models on several key metrics, including visual quality (SSIM, FID) and lip synchronization performance (Sync-C), demonstrating the `effectiveness and competitiveness` of this work. We will discuss the main issues below.
>
> > **The renderer is too central in metrics: The paper claims SOTA visual quality (SSIM, PSNR, FID) but uses a SOTA renderer (EchoMimic) as its final stage. This is a major confounding variable. These metrics are evaluating EchoMimic's rendering power, not just Text2Lip's landmark generation. The model's actual contribution (lip-sync/landmark quality) is not SOTA (Table 2).**
>
> Thank you for raising this insightful and valid concern. We would like to clarify that while EchoMimic is a powerful renderer, the reported visual quality metrics (SSIM, PSNR, FID) strongly attest to **the superior quality of the landmarks generated by Text2Lip**. Our reasoning is as follows:
>
> a) SSIM and PSNR are fundamental metrics measuring pixel-level and structural similarity. **No renderer, regardless of its capability, can output results with high pixel-wise alignment to real videos if the input structural guidance (e.g., facial contours, lip shapes) is inherently flawed**. The high SSIM and PSNR scores are therefore primarily attributable to the accurate structural guidance provided by our generated landmark sequences.
>
> b) FID assesses the distributional distance between generated and real images in a deep feature space. **Only if the landmark sequences from Text2Lip describe lip dynamics and facial motions that closely resemble real human speech will the final video distribution align with the real video distribution**. Thus, our best-in-class FID score demonstrates that the visual content defined by our landmarks is highly realistic and natural.
>
> c) To completely isolate the rendering step and directly evaluate our core innovation, i.e., **generating semantically accurate facial motion from text**. We would like to direct your attention to the semantic quality metrics (BLEU, WER) in `Tables 2 and 4`. These metrics are computed by performing "lip-reading" on the generated landmark sequences themselves, entirely bypassing the renderer. **Our significant lead in these metrics serves as the most direct evidence that the motion sequences generated by Text2Lip possess the highest semantic fidelity**.
>
> We acknowledge that our MPJPE in `Table 2` is not the best among compared methods. However, MPJPE is a geometric metric measuring joint coordinate error. **In talking face generation, semantic correctness is far more critical than millimeter-level geometric precision**. A landmark sequence with a slightly higher MPJPE is superior for the task if its motion patterns can be correctly recognized as the corresponding words. This is analogous to a lip shape being perceived as "correct" (high semantic quality) being far more valuable than one that numerically matches a reference frame (low MPJPE) but is unintelligible.
>
> > **Central Motivation is Unproven: The entire paper is based on solving "audio-to-lip ambiguity" (e.g., "bad boy" vs. "bat boat") leading to "blurred outputs." This claim is never proven. No experiment shows audio-driven models failing at this while Text2Lip succeeds.**
>
> We fully understand your concern regarding the need for more direct experimental evidence for our central motivation. One of our main considerations is grounded in a well-established challenge in audio-visual research: **the inherent many-to-one mapping between phonemes and visual articulation** (e.g., /p/, /b/, /m/ all corresponding to the same bilabial closure viseme). This audio-to-lip ambiguity can, in theory, lead models to learn an averaged and blurred mapping function. Our approach bypasses the audio-to-lip ambiguity that causes blurred outputs by mapping text to unambiguous visemes, systematically circumventing the limitations of audio-driven models.
>
> While we acknowledge that the paper does not present minimal-pair contrasting examples such as "bad boy" vs. "bat boat," the results in `Table 3` demonstrates that our text-only Text2Lip model significantly outperforms all audio-driven baselines on semantic metrics (BLEU/WER). Furthermore, the semantic quality results in the first two rows of `Table 4` illustrate that by introducing and utilizing visemes as a visually explicit intermediate representation, our method effectively bypasses the ambiguity encountered when directly inferring visual performance from either audio signals or raw text words.

---

> ### Author Response · Authors · 2025-11-27
> **Response to Reviewer yBU1 - Part2**
>
> > **Ignores Text-Only Limitations: A text-only input has no information about prosody, pace, or emotion. The claims of "micro-expression dynamics" are unsubstantiated, as the model likely generates flat, "average-toned" speech.**
>
> We agree that text-only input inherently lacks paralinguistic information about prosody, pace, or emotion, which constitutes a fundamental limitation of the text-driven paradigm. Accordingly, in the current work, we do not claim to generate talking faces with rich, controllable emotions or complex rhythmic patterns directly from text. Our original reference to "micro-expression dynamics" was intended to describe the natural facial dynamics related to specific articulatory movements (e.g., precise lip-shape changes triggered by certain phonemes and necessary facial muscle coordination), rather than the generation of high-level emotional expressions. These details are crucial for achieving realistic lip synchronization. To prevent any misunderstanding, we will revise the relevant description to "natural facial dynamics" in the revised manuscript to more accurately reflect the model's capabilities.
>
> That said, this limitation does not imply that our text-driven approach is inherently disadvantaged compared to audio-driven methods. **The core objective of this paper is to address talking face generation in scenarios where audio is unavailable**. In many practical applications, such as restoration of silent films or high-noise environments, the original audio is either absent or unusable, rendering audio-driven models entirely inapplicable. In such critical settings, our method provides **a viable and unique solution**. Our work thereby demonstrates the distinct value and irreplaceability of the text-driven paradigm under these specific but important conditions.
>
> > **Weak Generalization Claims: The model is benchmarked on small, outdated, closed-domain datasets (GRID, AVDigits). The model has likely memorized the limited vocabulary. The "unseen sentence" example (Fig. 5) appears to use words from the training corpus.**
>
> Thank you for this comment regarding generalization. We would like to clarify several key points regarding our experimental design and the demonstrated capabilities of Text2Lip.
>
> Regarding the choice of benchmark datasets, **GRID and AVDigits are well-established and widely adopted benchmarks in this research domain** (e.g., [R1-R5]). Our decision to use these datasets was strategic: their controlled settings (i.e., featuring single scenarios and frontal views) **allow us to minimize confounding factors** such as variable backgrounds, camera angles, and recording environments. This focused setup enables a precise evaluation of **the core capability of the text-driven paradigm to replace audio in guiding lip-synchronized video generation**. While these datasets have a constrained vocabulary, this very limitation provides an ideal testbed for validating the feasibility and effectiveness of the novel cross-modal mapping from text to visemes to visual articulation that we propose. Our results demonstrate that even within this bounded lexicon, our text-driven approach achieves performance comparable to audio-driven methods, thereby establishing a solid foundation for future exploration on larger-scale, open-domain datasets.
>
> Concerning the evaluation on "unseen sentences" in `Fig. 5`, we wish to emphasize that **using words from the training vocabulary to form novel, previously unseen sentences is a standard and core paradigm for assessing compositional generalization**. The model's task is not to memorize specific sentences but to learn a generalizable mapping from text to facial motion. When presented with a new sentence composed of known words in a novel syntactic and semantic sequence, the model must **dynamically interpret its meaning and correctly compose and sequence the learned phoneme-to-viseme mappings** to generate coherent and accurate lip movements. Its success in this task provides direct evidence of robust compositional generalization. Furthermore, our cross-speaker tests (`Fig. 4`) show that the model successfully transfers learned articulatory patterns to new facial structures and identities, validating its strong generalization capability across speakers. Together, these results indicate that our model has learned underlying articulatory principles rather than merely memorizing the training data.
>
> - [R1] On the Audio-visual Synchronization for Lip-to-speech Synthesis. CVPR, 2023.
> - [R2] Speech2lip: High-fidelity Speech to Lip Generation by Learning from A Short Video. CVPR, 2023.
> - [R3] Automatic Lip-reading Classification using Deep Learning Approaches and Optimized Quaternion Meixner Moments by GWO Algorithm. KBS, 2024.
> - [R4] CFLip: Generalizing Lipreading to Unseen Speakers by Learning Common Features. IEEE TCSS, 2025.
> - [R5] Realistic Real-Time Talking Head Synthesis with Grid Encoding and Progressive Conditioning. ICASSP, 2025.

---

> ### Author Response · Authors · 2025-11-27
> **Response to Reviewer yBU1 - Part3**
>
> > **Questionable Comparisons: The landmark generation (Table 2) is compared against Sign Language Production models, which is not explained.**
>
> Regarding the comparison with sign language production models in `Table 2`, this experimental design is motivated by two key considerations:
>
> From a technical perspective, sign language production and our lip landmark generation face the same fundamental challenge: how to accurately map discrete linguistic symbols into continuous, precise human motion trajectories. Although the application domains differ, both tasks require solving the core problem of **language-to-motion mapping**. The sign language production field, having undergone longer development, has established robust baseline models such as PT-GN and GEN-OBT, which provide reliable benchmarks for evaluating "language-motion" mapping capabilities.
>
> Besides, by employing task-agnostic semantic fidelity metrics (BLEU/WER), we can **isolate the influence of specific application domains** and directly evaluate different models' capability in learning and executing the fundamental "language-motion" mapping task. The fact that our method outperforms specialized sign language production models on these universal metrics precisely demonstrates the effectiveness and generalization capability of our text-to-viseme mapping mechanism in addressing cross-modal generation challenges.
>
> > **Contradictory Justification: The paper fails to mention that visemes also suffer from ambiguity (many different phonemes/sounds map to the same viseme), which undermines its own central premise.**
>
> We wish to clarify that the key linguistic phenomenon of "many different phonemes/sounds map to the same viseme" **has been explicitly addressed in our submission**, both in `Fig. 2` and in `lines 141-147` of the main text.
>
> This phenomenon actually validates the core motivation of our approach: **the inherent many-to-one mapping from phonemes to visual articulation represents the fundamental challenge for audio-driven methods**. Visemes, as basic units of visual speech, provide a standardized visual abstraction for this mapping. By introducing visemes as an intermediate representation, we transform the problem **from an uncertain "audio-to-visual" mapping to a deterministic "text-to-viseme-to-visual" mapping**. Rather than forcing the model to deduce visual output from ambiguous audio signals, our approach enables direct learning from deterministic viseme sequences to deterministic visual representations. This design not only doesn't undermine our premise, but rather confirms **the value of visemes as visual intermediaries in resolving phoneme-visual ambiguity**.
>
> > **Q1 (The Renderer): Why are visual quality metrics (SSIM, FID) better than the EchoMimic baseline itself (Table 1)? Since Text2Lip uses EchoMimic, this result seems contradictory. Does this imply the generated landmarks are "better than real" in some way that improves rendering?**
>
> We appreciate the reviewer's insightful question. We would like to clarify a crucial experimental detail that was not sufficiently explained in the manuscript. In `Table 1`, all baseline models (including EchoMimic) were evaluated using their original architectures where audio serves as the mandatory core input signal, while text or landmark inputs remain optional auxiliary modalities. To ensure a fair comparison of single-modality driving capabilities, the reported baseline results correspond to configurations using only audio input. This establishes a direct and meaningful comparison with our Text2Lip approach, which operates exclusively on text input.
>
> This comparative outcome precisely demonstrates the core value of our method: by completely eliminating reliance on audio input and leveraging our text-driven viseme-landmark mapping, we achieve visual quality that matches or even surpasses traditional audio-driven approaches. We will explicitly clarify this comparative setup in the revised manuscript to prevent any potential misunderstanding.

---

> ### Author Response · Authors · 2025-11-27
> **Response to Reviewer yBU1 - Part4**
>
> > **Q2 (The Ambiguity Claim): Why do the authors claim identical lip shapes would lead to "blurred outputs"? A model should just learn to output the identical shape. Where is the evidence that this ambiguity is a real problem that audio-models fail on?**
>
> We appreciate the reviewer's insightful question regarding our claim about ambiguous audio-lip mapping leading to blurred outputs. We understand the theoretical expectation that a model should learn to produce sharp outputs for identical target shapes. However, there exists a crucial discrepancy between this theoretical perspective and the practical optimization behavior of real-world models, which is the root of the misunderstanding.
>
> **The core issue lies in the fundamental nature of the "audio-to-lip" mapping: it is not a deterministic one-to-one function but an ambiguous many-to-one process**. When facing different audio inputs (e.g., the audio streams of "bad boy" vs. "bat boat"), the ideal target outputs are indeed similar yet distinct, sharp lip sequences. However, since these subtle differences are challenging to capture precisely from ambiguous audio features, the model faces an optimization dilemma during training: **it cannot deterministically learn to match each subtly different audio input with its unique correct visual output**. Consequently, as a "shortcut," the model tends to learn an averaged mapping—producing a visually compromised result that represents a compromise between all perceptually similar inputs. This compromise manifests as blurred outputs in image space, not because the model lacks the capacity to generate sharp images, but because the inherent uncertainty in the learned mapping function propagates from input space to output space.
>
> The semantic quality metrics (BLEU/WER) in our `Table 2` provide compelling indirect evidence for this phenomenon. If audio-driven models could perfectly resolve this ambiguity, their generated lip sequences should achieve the highest semantic clarity. However, the fact that our text-only model significantly outperforms them on these metrics strongly suggests that audio-driven models inherently struggle to recover accurate visual semantics from ambiguous audio signals, resulting in semantically "blurred" outputs that ultimately underlie the observed visual blurriness.
>
> > **Q3 (The Application): If the model relies only on text, how is synchronicity with a separate audio track (for a final dubbed video) ensured? Does this not require a second, separate model or manual alignment, defeating the purpose?**
>
> The question rightly stems from a reasonable application assumption: the need to synchronize with a separate, pre-recorded audio track. However, this precisely highlights the fundamental difference in approach between our method and traditional audio-driven paradigms.
>
> **Our core solution lies in using text as a unified timing source to achieve endogenous synchronization of audiovisual signals**. Specifically, when text is input, the model simultaneously generates two aligned outputs: precise facial motion sequences and pseudo-acoustic features derived from text semantics. These two outputs share the same text-based temporal logic during generation, enabling the final rendered video to possess inherent audiovisual synchronization. This approach is fundamentally designed not for "dubbing pre-existing audio" but for **creating complete audiovisual content from scratch**. This design gives our method unique advantages in scenarios such as virtual avatar driving and instant video content generation—where **text serves as the starting point for creation rather than a post-hoc addition**. This is not a limitation of our method but rather defines its distinct application boundaries compared to traditional audio-driven paradigms, reflecting its original design purpose.
>
> > **Q4 (The Ablations): Why do the ablation studies (Table 4) show improvements in image quality metrics (SSIM, FID)? The renderer is pre-trained and fixed. This implies the landmark quality alone is responsible, but this is not measured with a temporal metric like FVD, which would be more appropriate.**
>
> Thank you for this insightful observation. The reviewer correctly notes that with a fixed renderer, improvements in image quality metrics must be attributed to enhanced landmark quality. We fully acknowledge the importance of temporal evaluation metrics and have validated this through Frèchet Video Distance (FVD) measurements. As shown in the table below, Text2Lip achieves an FVD score of 762.683, demonstrating significant superiority over both the "Text only" and "Text→Viseme" baselines.
>
> This finding, together with improvements in SSIM and FID, confirms the crucial role of landmark quality enhancement. We will enhance this analysis in the revision.
>
> | Methods | FVD |
> |--------------|--------------|
> |Text | 3912.511 |
> |Text→Viseme | 2673.519 |
> |Text2Lip(Ours) | 762.683 |

---

### Author Response · Authors · 2025-12-04
**Rebuttal Summary for the Area Chair**

Here is a summary of the key points from the reviews and our responses, to provide clarity and demonstrate that all major concerns have been adequately addressed.

## Core Contribution of Our Work
**Text2Lip**, introduces a novel paradigm for **text-driven lip-synced talking face generation**. Its primary contribution is threefold:

- **A Viseme-Centric Framework**: It bridges linguistic semantics and visual articulation by converting text into structured **viseme sequences**, providing a linguistically grounded prior that resolves the inherent ambiguity in audio-to-lip mapping.

- **Progressive Viseme-Audio Replacement (PVAR)**: A curriculum learning strategy that progressively replaces real audio with viseme-derived features during training. This enables robust **audio-free generation** while maintaining the ability to function with audio when available.

- **Modality-Robust Synthesis**: The method demonstrates that high-quality, synchronized talking faces can be generated **using text alone**, challenging the prevailing assumption that precise lip synchronization necessitates direct audio input. It establishes a new, flexible paradigm for scenarios where audio is unavailable, noisy, or privacy-sensitive.

## Key Strengths Highlighted by Reviewers
Reviewers yBU1, yBxW, tDSp, and Aj3c unanimously acknowledged several **strengths**:

- **Novelty and Clear Motivation**: Addressing the "audio-to-lip ambiguity" problem and the scarcity of high-quality audio-visual data.

- **Effective Training Strategy**: The PVAR strategy was praised as technically straightforward, well-motivated, and effective for achieving text-only inference.

- **Strong Empirical Performance**: Our method achieves **state-of-the-art or highly competitive results** on standard benchmarks across multiple metrics: visual quality, semantic fidelity, and lip synchronization.

- **Comprehensive Evaluation**: The experimental section was commended for its multi-axis evaluation covering visual fidelity, synchronization, landmark accuracy, and semantic quality.

## Responses to Major Concerns

- **Evidence for Core Motivation**: We clarified that the **superior semantic fidelity (BLEU/WER) of our text-only model over all audio-driven baselines** provides strong evidence. This result demonstrates that our viseme-based approach effectively bypasses the ambiguity that hinders audio-driven models, as poor semantic scores would indicate "blurred" semantic mapping.

- **Evaluation Fairness and Confounding Variables**:

> **Renderer (EchoMimic)**: We clarified that in Table 1, the **EchoMimic baseline uses only audio**, while Text2Lip uses only text, making the comparison fair. The superior visual metrics stem from our model's higher-quality landmark sequences, which provide better structural guidance to the same renderer. We further supported this with new **FVD (Fréchet Video Distance) results** (Text2Lip: 762.683 vs. baselines >2600), proving our landmarks also yield superior temporal coherence.

> **SyncNet Protocol**: We explicitly stated that **all Sync-C/D scores are computed using the dataset's ground-truth audio**, never pseudo-audio. This ensures a fair comparison and, in fact, poses a greater challenge for our audio-free model. Its competitive scores under this condition strongly validate the quality of our generated motion.

- **Methodological Design**:

> **Pipeline Complexity**: Our pipeline is **not a cascade of five independent stages** but a streamlined two-stage process: (1) an **end-to-end text-to-landmark module** (where "pseudo-audio" is an internal latent feature), and (2) a rendering stage. This decoupling allows focused optimization and better generalization.

> **PVAR Strategy**: The **ablation studies** showing that our progressive curriculum strategy **significantly outperforms fixed-probability random dropout** across key metrics, justifying its design for stable convergence to a "zero-audio" state.

> **Duration & Coarticulation**: We detailed the heuristic and implicit mechanisms for handling viseme durations and transitions. The model's context-aware viseme encoder and continuous landmark decoder inherently learn to model **coarticulation and smooth transitions**.

- **Scope and Generalization**:

> **Datasets**: We justified the use of standard, controlled benchmarks for **fair, comparable evaluation** and showed preliminary cross-speaker, cross-style, and novel-sentence results indicating generalization beyond memorization.

> **Emotional Expression**: We acknowledged this as a **fundamental limitation of text-only input** and reframed our contribution: we solve the **core challenge of accurate lip sync in audio-absent scenarios**, where audio-driven models fail entirely.

> **NSLT for Evaluation**: We justified its use due to **task similarity** (keypoint sequence to semantics) and its advantage in **isolating landmark quality from rendering artifacts**. We confirmed its validity via benchmarking on ground-truth data.

---

### Meta-Review · Area_Chair_8oKf · 2026-01-06

**Summary:**

This paper explores a paradigm of text-to-lip generation by introducing the Progressive Viseme-Audio Replacement (PVAR) mechanism to construct viseme-based linguistic priors, aiming to address the inherent challenges of data dependency and mapping ambiguity found in traditional audio-driven methods. The paper received initial scores of 2/6/6/4, and remained unchanged before the reset.

The reviewers’ initial concerns mainly centered on the empirical support for the core motivation, the fairness of the evaluation protocol, the functional limitations of text-only input, and the lack of viseme duration modeling. Prior to the score reset, the authors addressed most of these concerns by using the advantages of semantic metrics (BLEU/WER) to verify the effectiveness in resolving audio-to-lip ambiguity, clarifying that Sync-C/D metrics are uniformly calculated using real audio, revising the description of the capability boundaries of the text-driven paradigm, and supplementing a heuristic duration estimation method based on training data statistics.

However, the authors’ response fails to convincingly address several critical concerns. Specifically, reviewer yBU1's observation regarding the loss of emotional expressivity and prosody in text-only inputs remains unaddressed, which in turn leaves reviewer Aj3c’s concerns about the lack of human-like naturalness unresolved. Furthermore, the rebuttal does not include the additional experiments on diverse datasets requested by both reviewers, leaving the model’s generalization capabilities insufficiently validated.

After a careful assessment of the submission, reviews, response, and discussion, the AC recommends rejection. The authors are encouraged to revise and refine the manuscript in accordance with the reviewers’ feedback for a future submission.

**Reviewer Concerns:**

The reviewers’ initial concerns mainly centered on the empirical support for the core motivation, the fairness of the evaluation protocol, the functional limitations of text-only input, and the lack of viseme duration modeling. Prior to the score reset, the authors addressed most of these concerns by using the advantages of semantic metrics (BLEU/WER) to verify the effectiveness in resolving audio-to-lip ambiguity, clarifying that Sync-C/D metrics are uniformly calculated using real audio, revising the description of the capability boundaries of the text-driven paradigm, and supplementing a heuristic duration estimation method based on training data statistics.

However, the authors’ response fails to convincingly address several critical concerns. Specifically, reviewer yBU1's observation regarding the loss of emotional expressivity and prosody in text-only inputs remains unaddressed, which in turn leaves reviewer Aj3c’s concerns about the lack of human-like naturalness unresolved. Furthermore, the rebuttal does not include the additional experiments on diverse datasets requested by both reviewers, leaving the model’s generalization capabilities insufficiently validated.

**Reviewer Scores:**

The manuscript received initial review scores of 2/6/6/4. After the rebuttal/discussion and before the reset, the score remained unchanged.

Since several concerns raised by the reviewers may remain unresolved after the rebuttal (see 'Reviewer Concerns'), I would approximate 2/6/6/4 as the final score.

---

### Decision · Program_Chairs · 2026-01-26

Reject